# Synergy Between Sufficient Changes and Sparse Mixing Procedure for Disentangled Representation Learning

**Zijian Li**[†••*] **Shunxing Fan**[••*] **Yujia Zheng**[†] **Ignavier Ng**[†] **Shaoan Xie**[†] **Guangyi Chen**[†•]
**Xinshuai Dong**[†] **Ruichu Cai**[‡] **Kun Zhang**[†•]
[†]Carnegie Mellon University, Pittsburgh PA, USA
[•]Mohamed bin Zayed University of Artificial Intelligence, Abu Dhabi, UAE
[‡]Guangdong University of Technology, Guangzhou, China

## Abstract

Disentangled representation learning aims to uncover latent variables underlying the observed data, and generally speaking, rather strong assumptions are needed to ensure identifiability. Some approaches rely on sufficient changes on the distribution of latent variables indicated by auxiliary variables such as domain indices, but acquiring enough domains is often challenging. Alternative approaches exploit structural sparsity assumptions on the mixing procedure, but such constraints are usually (partially) violated in practice. Interestingly, we find that these two seemingly unrelated assumptions can actually complement each other to achieve identifiability. Specifically, when conditioned on auxiliary variables, the sparse mixing procedure assumption provides structural constraints on the mapping from estimated to true latent variables and hence compensates for potentially insufficient distribution changes. Building on this insight, we propose an identifiability theory with less restrictive constraints regarding distribution changes and the sparse mixing procedure, enhancing applicability to real-world scenarios. Additionally, we develop an estimation framework incorporating a domain encoding network and a sparse mixing constraint and provide two implementations, based on variational autoencoders and generative adversarial networks, respectively. Experiment results on synthetic and real-world datasets support our theoretical results. The code is available at https://github.com/jozerozero/Synergy_Disentanglement.

## 1 Introduction

Disentangled representation learning (Schölkopf et al., 2021) is a learning paradigm that seeks to learn meaningful and explanatory representation from observed data. Mathematically, suppose that observed variables $\mathbf{x}$ are generated from latent variables $\mathbf{z}$ through an unknown mixing function $f$, i.e., $\mathbf{x} = f(\mathbf{z})$. The primary goal of disentangled representation learning is to recover the latent variables with certain identifiability guarantees, i.e., to estimate them up to particular indeterminacies such as component-wise transformations. A theoretical foundation in this field is related to independent component analysis (ICA) (Hyvärinen et al., 2001; 2023). Earlier methods (Hyvärinen & Oja, 2000; Tichavsky et al., 2006) achieved identifiability with the linear mixing assumption. To address more complex real-world tasks, nonlinear ICA (Hyvarinen & Morioka, 2016) allows the mixing function $f$ to be an unknown nonlinear function. However, the identifiability of nonlinear ICA remains a significant challenge, primarily because without additional assumptions, there can be infinitely many solutions with independent variables which are mixtures of the true latent variables (Hyvärinen & Pajunen, 1999).

To overcome these challenges, several assumptions have been proposed, such as distribution sufficient changes (see, e.g., Hyvarinen & Morioka (2016)) and mixing procedure restrictions (see, e.g., Zheng & Zhang (2023)). For example, the methods based on sufficient changes exploit the auxiliary valuables $\mathbf{u}$ and assume that the latent variables have changing distributions across different values

---

*These authors contributed equally to this work.

of $\mathbf{u}$ but are conditionally independent given $\mathbf{u}$ (Hyvarinen & Morioka, 2016; Hyvarinen et al., 2019; Khemakhem et al., 2020a). Under this assumption, many works use domain indices or the observed class labels as $\mathbf{u}$(Kong et al., 2022; Li et al., 2023). Additionally, other works harness the clustering-based methods (Willetts & Paige, 2021) or temporal dependency (Yao et al., 2021; 2022; Chen et al., 2024; Li et al., 2024) for the identifiability of nonlinear ICA. Alternatively, one can impose suitable restrictions on the mixing function (Gresele et al., 2021; Moran et al., 2021; Locatello et al., 2019b) to achieve identifiability. Specifically, Hyvärinen & Pajunen (1999); Buchholz et al. (2022) show that conformal maps are identifiable up to specific indeterminacies. Recently, Morioka & Hyvärinen (2023) achieved identifiability by assuming that the observational mixing exhibits a suitable grouping of the observations. Zheng et al. (2022); Zheng & Zhang (2023) proposed the structural sparsity assumption, making it a fully unsupervised manner for the identifiability of nonlinear ICA. Specifically, this assumption restricts the connective structure from sources to observations, i.e., the support of the Jacobian matrix of the mixing function. One may refer to Appendix C for further discussion of related works, including the disentangled representation learning, identifiability of nonlinear ICA, and multi-domain image generation.

Despite advances in the identifiability of disentangled representation, these methods impose stringent conditions on the number of auxiliary variables or the mixing function, each of which might be violated in practice. On the one hand, the methods based on auxiliary variables Khemakhem et al. (2020a); Kong et al. (2022) often require a large number of values of the auxiliary variables $\mathbf{u}$. Specifically, in domain adaptation, achieving theoretical guarantees of component-wise identifiability of the latent variables typically requires at least $2n + 1$ domains, where the dimension of latent variables is $n$. On the other hand, to achieve full identifiability, methods with a constrained mixing procedure have to enforce strong constraints on the mixing function. For example, the structural sparsity assumptions (Zheng et al., 2022; Zheng & Zhang, 2023)say that for each latent source $z_i$, there exists a set of observations such that the intersection of their parents is $z_i$, which excludes scenarios where any latent variable is densely connected to observed variables. Although each of the two types of constraints discussed above is generally strong, both of them may be partly true in practice, given a particular problem. Therefore, it is natural to ask this question: *Is it possible to leverage both types of constraints in a complementary, principled way to learn disentangled representations with identifiability guarantees?*

The answer is **yes**–a unified framework is proposed in this paper to address it. Intuitively, the sparse mixing procedure assumption implies independence between certain latent variables and a particular subset of the observed variables. For example, $z_i \perp x_j | \mathbf{u}$, if $z_i$ is not adjacent to $x_j$ (i.e., $z_i$ does not contribute to $x_j$). This condition can then be used to constrain the mapping from the estimated latent variables $\hat{\mathbf{z}}$ to the true latent variables $\mathbf{z}$. In such cases, the sparsity in the mixing procedure compensates for a potential lack of sufficient domains. Additionally, even when the structural sparsity assumption is partially violated, the sufficient changes introduced by the auxiliary variables $\mathbf{u}$ can still benefit the identifiability of latent variables. Therefore, the sparse mixing procedure and sufficient changes assumptions nicely complement each other—when one assumption is (partially) violated, the other can compensate, allowing for the identifiability of disentangled representations with milder assumptions.

Based on this intuition, we establish a principled identifiability framework leveraging both types of constraints. Moreover, we develop a general generative model framework with domain encoding networks and a sparse mixing constraint for disentangled representation learning benefiting from both constraints. Specifically, the domain encoding network is used to impose the assumption of sufficient changes by modeling the distribution of latent variables given auxiliary variables, and the sparse mixing constraint is used to enforce the sparsity of the estimated mixing procedure. Our method is validated through a simulation and several widely used multi-domain image generation datasets. The performance demonstrates the effectiveness of the proposed framework.

## 2 PRELIMINARIES

We start by giving a brief background on nonlinear independence component analysis (ICA) and identifiability. In a typical nonlinear ICA setting, the data generation process is shown as follows:

$$p(\mathbf{z}) = \prod_{i=1}^{n} p(z_i), \quad \mathbf{x} = g(\mathbf{z}), \tag{1}$$

Table 1: Notation Descriptions.

| Symbol | Description |
| --- | --- |
| $z$ | Scalar variable |
| $\mathbf{x}$ | Random vector |
| $\mathbf{x}_k$ | $k$-th component of the random vector $\mathbf{x}$ |
| $\mathbf{x}_{(i)}$ | $i$-th instance or realization of the random vector $\mathbf{x}$ |
| $\mathbf{x}_{k,(i)}$ | $i$-th instance of the $k$-th component of the random vector $\mathbf{x}$ |
| $\mathbf{x}_{\setminus k}$ | Random vector $\mathbf{x}$ excluding the $k$-th component |

where $\mathbf{x} = (x_1, \ldots, x_n)$ and $\mathbf{z} = (z_1, \ldots, z_n)$ denote the observed variables and the independent latent variables with the dimension of $n$, respectively, function $g : \mathbf{z} \to \mathbf{x}$ denotes a nonlinear mixing procedure. As mentioned in Hyvärinen & Pajunen (1999), without any further assumptions, there is an infinite number of possible solutions and these have no trivial relation with each other. Therefore, previous works introduce the auxiliary variables $\mathbf{u}$ and assume that the latent variables are conditionally independent given $\mathbf{u}$., i.e., $\mathbf{z} = f_u(\epsilon)$ and $p(\mathbf{z}|\mathbf{u}) = \prod_{i=1}^n p(z_i|\mathbf{u})$, where $\epsilon$ is the independent exogenous noise variables. The goal of nonlinear ICA is to learn an estimated unmixing function $\hat{g}^{-1} : \mathbf{x} \to \hat{\mathbf{z}}$, such that $\hat{\mathbf{z}} = (\hat{z}_1, \ldots, \hat{z}_n)$ consists of independent estimated sources.

To better understand our theoretical results, we provide the description of notation as shown in Table 1 as well as the definition of subspace-wise identifiability and component-wise identifiability.

**Definition 1** (**Subspace-wise Identifiability of Latent Variables** (Li et al., 2023)). *The subspace-wise identifiability of $\mathbf{z} \in \mathbb{R}^{n_d}$ means that for ground-truth $\mathbf{z}$, there exists $\hat{\mathbf{z}}$ and an invertible function $h : \mathbb{R}^{n_d} \to \mathbb{R}^{n_d}$, such that $\mathbf{z} = h(\hat{\mathbf{z}})$.*

**Definition 2** (**Component-wise Identifiability of Latent Variables** (Hyvarinen & Morioka, 2016)). *The component-wise identifiability of $\mathbf{z}$ is that for each $z_i, i \in [n]$, there exists a corresponding estimated component $\hat{z}_j, j \in [n]$ and an invertible function $h_i : \mathbb{R} \to \mathbb{R}$, such that $z_i = h(\hat{z}_j)$.*

## 3 IDENTIFIABILITY WITH COMPLEMENTARY GAINS FROM SUFFICIENT CHANGES AND SPARSE MIXING PROCEDURE

In this section, we illustrate the identifiability results by leveraging the complementary gains from sufficient changes and sparse mixing procedures. Specifically, we first present the subspace identification result (**Theorem** 1), where certain latent variables are subspace-wise identification. Based on the aforementioned result, we further establish the component-wise identification result (**Theorem** 2), which uses conditional independence induced by sparse mixing procedure to constraint the solution space of a full-rank linear system. Additionally, we also show that the existing component-wise identification results (Khemakhem et al., 2020a; Kong et al., 2022) with $2n + 1$ auxiliary variables are special cases of our approach (**Corollary** 1) when the mixing procedure from latent sources to observations are fully-connected.

### 3.1 SUBSPACE IDENTIFIABILITY WITH COMPLEMENTARY GAINS

**Theorem 1.** *(**Subspace Identification with Complementary Gains**) Following the data generation process in Equation (1), we further make the following assumption.*

- **A1 (Smooth and Positive Density)**: *The probability density function of latent variables is smooth and positive, i.e., $p_{\mathbf{z}|\mathbf{u}} > 0$ over $\mathcal{Z}$ and $\mathcal{U}$.*

- **A2 (Conditional Independence)**: *Conditioned on $\mathbf{u}$, each $z_i$ is independent of any other $z_j$ for $i, j \in \{1, \ldots, n\}, i \neq j$, i.e., $\log p_{\mathbf{z}|\mathbf{u}}(\mathbf{z}|\mathbf{u}) = \sum_{i=1}^n \log p_{z_i|\mathbf{u}}(z_i|\mathbf{u})$.*

- **A3 (Generalized Sufficient Changes for Subspace Identification)** *Let $\mathbf{x}_k$ be a subset of $\mathbf{x}$, $\mathbf{x}_{\setminus k}$ be the left variables, and $\mathbf{x}_{k,(1)}, \mathbf{x}_{k,(0)}$ be two different instance of $\mathbf{x}_k$. And vectors $w(k, \mathbf{u}) - w(k, 0)$ with $\mathbf{u} = 1, \ldots, |Pa(\mathbf{x}_k)|$ are linearly independent, where vector $w(k, \mathbf{u})$ is defined as:*

$$
\begin{aligned}
w(k, \mathbf{u}) = \Big( & \frac{\partial \log p(\mathbf{x}_{k,(1)}, \mathbf{x}_{\setminus k}|\mathbf{u})}{\partial z_1}, -\frac{\partial \log p(\mathbf{x}_{k,(0)}, \mathbf{x}_{\setminus k}|\mathbf{u})}{\partial z_1}, \ldots, \\
& \frac{\partial \log p(\mathbf{x}_{k,(1)}, \mathbf{x}_{\setminus k}|\mathbf{u})}{\partial z_{Pa(\mathbf{x}_k)}}, -\frac{\partial \log p(\mathbf{x}_{k,(0)}, \mathbf{x}_{\setminus k}|\mathbf{u})}{\partial z_{Pa(\mathbf{x}_k)}} \Big).
\end{aligned}
\tag{2}
$$

*Suppose that we learn $\hat{g}$ to achieve Equation (1) with the minimal number of edges of the mixing process. Then, for every pair of $\hat{z}_j$ and $\mathbf{x}_k$ in which $\mathbf{x}_k$ does not contribute to $\hat{z}_j$, i.e., $\frac{\partial \hat{\mathbf{z}}_j}{\partial \mathbf{x}_k} \equiv 0$, $\mathbf{z}_{Pa(\mathbf{x}_k)}$ is not the function of $\hat{z}_j$, i.e., $\frac{\partial \mathbf{z}_{Pa(\mathbf{x}_k)}}{\partial \hat{z}_j} \equiv 0$, where $\mathbf{z}_{Pa(\mathbf{x}_k)}$ is the parents of $\mathbf{x}_k$*

**Intuition of the Theoretical Results.** A proof of Theorem 1 can be found in Appendix A.1. For a better understanding of the proposed theory, we provide an intuitive example shown as Figure 1, which includes the ground truth and estimated data generation processes with auxiliary variables $\mathbf{u}$. The solid lines denote the true mixing edges. In the ground-truth generation process, because $\mathbf{z}_a$ is not adjacent to $\mathbf{x}_k$, $\frac{\partial \mathbf{x}_k}{\partial \mathbf{z}_a} \equiv 0$. Since the powerful neural networks may choose to use the fully connected mixing procedure during training, leading to the redundant estimated mixing edges, i.e., the red dashed line. We will show how to remove these redundant estimated mixing edges via sparse mixing constraints in Section 4.

As shown in Figure 1 (a), we let $\mathbf{x}_k$ and $\mathbf{z}_k$ be a subset of observed variables and its corresponding parents. Since $\mathbf{z}_a$ do not connect to $\mathbf{x}_k$, we have $\mathbf{x}_k \perp\!\!\!\perp \mathbf{z}_a | \mathbf{u}$. Sequentially, suppose that we have two different values of $\mathbf{x}_k$, i.e., $\mathbf{x}_{k,(0)}$ and $\mathbf{x}_{k,(1)}$, then we have:

$$\frac{\partial \log p(\mathbf{x}_a, \mathbf{x}_b, \mathbf{x}_{k,(1)} | \mathbf{u})}{\partial \mathbf{z}_a} - \frac{\partial \log p(\mathbf{x}_a, \mathbf{x}_b, \mathbf{x}_{k,(0)} | \mathbf{u})}{\partial \mathbf{z}_a} = 0. \tag{3}$$

Similarly, by constraining the sparsity of the estimated mixing procedure, we can also achieve conditional independence $\hat{\mathbf{z}}_a \perp\!\!\!\perp \hat{\mathbf{x}}_k | \mathbf{u}$. Therefore, we can construct a full-rank linear system as shown in Equation (4) by leveraging auxiliary variables.

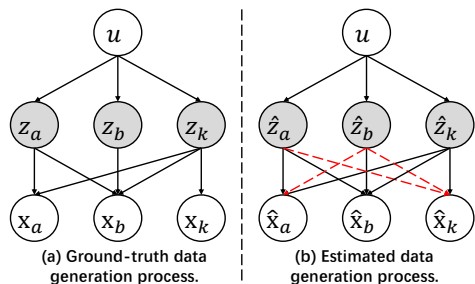

(a) Ground-truth data generation process.

(b) Estimated data generation process.

Figure 1: Example for Theorem 1 with ground-truth and estimated data generation processes. The red dashed lines denote the redundant estimated mixing edges.

$$\sum_{z_i \in \mathbf{z}_k} \left( \frac{\partial \log p(\mathbf{x}_a, \mathbf{x}_b, \mathbf{x}_{k,(1)} | \mathbf{u})}{\partial z_i} \cdot \frac{\partial z_i}{\partial \hat{z}_j} \Big|_{\mathbf{x}_k = \mathbf{x}_{k,(1)}} - \frac{\partial \log p(\mathbf{x}_a, \mathbf{x}_b, \mathbf{x}_{k,(0)} | \mathbf{u})}{\partial z_i} \cdot \frac{\partial z_i}{\partial \hat{z}_j} \Big|_{\mathbf{x}_k = \mathbf{x}_{k,(0)}} \right) = 0. \tag{4}$$

Therefore, with a sufficient number of values of the auxiliary variables $\mathbf{u}$, the only solution of $\frac{\partial \mathbf{z}_i}{\partial \hat{\mathbf{z}}_j}$, for $z_i \in \mathbf{z}_k$ and $\hat{z}_j \in \{\hat{\mathbf{z}}_a, \hat{\mathbf{z}}_b\}$, is zero. This implies that $\{\mathbf{z}_a, \mathbf{z}_b\}$ is only the function of $\{\hat{\mathbf{z}}_a, \hat{\mathbf{z}}_b\}$, i.e., $\{\mathbf{z}_a, \mathbf{z}_b\}$ is subspace-wise identifiable.

**Discussion of Assumptions.** We also provide detailed discussions of the assumptions and how they relate to real-world scenarios. The first two assumptions are standard in the componentwise identification of existing nonlinear ICA (Li et al., 2023; Kong et al., 2022; Khemakhem et al., 2020a). The smooth and positive density assumption means that the latent variables change continuously. For example, $\mathbf{z}_i$ can be considered as the light directions of images, which are changing continuously. The conditional independence assumption implies that there are no causal relationships among $\mathbf{z}$. For instance, considering gender as an auxiliary variable, the light direction and the image resolution are conditionally independent. Finally, the sufficient changes assumption reflects that the conditional distributions are first-order differentiable. And the linear independence is the condition of a unique solution of the linear full-rank system.

Although existing works like domain adaptation (Li et al., 2023), and disentangled representation learning (Zheng & Zhang, 2023) also adopt the linear independence of first-order to achieve subspace identification, the assumption in our work is more general and easier to meet in practice. First, the aforementioned methods achieve subspace identifiability by assuming that the auxiliary variables do not have influence on all the latent variables, e.g., partition the latent variables into domain-invariant and domain-specific variables. However, our result does not need this assumption. Moreover, the aforementioned methods only use an adequate number of values of the auxiliary variables to meet the linear independence assumption, which might be hard to satisfy in practice. Meanwhile, our method further harnesses the independence between observed and latent variables, which constraints the solution space of the full-rank linear system and hence decreases the requirement for the number of auxiliary variables.

## 3.2 COMPONENT-WISE IDENTIFIABILITY WITH COMPLEMENTARY GAINS

**Theorem 2.** *(**Component-wise Identification with Complementary Gains**) Let the observations be sampled from the data generating process in Equation (1). Suppose that Assumptions from Theorem 1 hold and also that we learn $\hat{g}, p_{\hat{\mathbf{z}}|\mathbf{u}}$ to match the marginal distribution between $\mathbf{x}$ and $\hat{\mathbf{x}}$ with the minimal number of edges of the mixing process. We further make the following assumption:*

- *A4 (**Sufficient Changes for Component-wise Identification.**): Suppose $\mathbf{x}_k$ and $\mathbf{x}_a$ be a subset of $\mathbf{x}$ and $\mathbf{x}_k \cap \mathbf{x}_a = \emptyset$. We let the $\mathbf{z}_a$ be the set of latent variables that connect to $\mathbf{x}_a$ but not connect to $\mathbf{x}_k$, such that for all different values of $\mathbf{x}_a$ such as $\mathbf{x}_{a,0}, \mathbf{x}_{a,(1)}$, the vectors $\mathcal{V}(a, k, \mathbf{u}_s) - \mathcal{V}(a, k, 0)$ with $s = 1, \ldots, 2|\mathbf{z}_a|$ are linearly independent. And the vector $\mathcal{V}(a, k, \mathbf{u}_s)$ is defined as follows:*

$$
\begin{aligned}
\mathcal{V}(a, k, \mathbf{u}_m) = \Big( & \frac{\partial^2 \log p(\mathbf{z}_{a,(1)}, \mathbf{z}_b, \mathbf{z}_k | \mathbf{u}_m)}{(\partial z_i)^2}, -\frac{\partial^2 \log p(\mathbf{z}_{a,(0)}, \mathbf{z}_b, \mathbf{z}_k | \mathbf{u}_m)}{(\partial z_i)^2}, \ldots, \\
& \frac{\partial \log p(\mathbf{z}_{a,(1)}, \mathbf{z}_b, \mathbf{z}_k | \mathbf{u}_m)}{\partial z_i}, -\frac{\partial \log p(\mathbf{z}_{a,(0)}, \mathbf{z}_b, \mathbf{z}_k | \mathbf{u}_m)}{\partial z_i} \Big)_{z_i \in \mathbf{z}_a}.
\end{aligned}
\tag{5}
$$

*Suppose that we learn $\hat{g}$ to achieve Equation (1) with the minimal number of edges of the mixing process. Then $\mathbf{z}_a$ is component-wise identifiable.*

**Intuition of the Theoretical Results.** A proof can be found in Appendix A.2. For a better understanding, we also provide an example as shown in Figure 1. Besides assuming that $\mathbf{x}_k$ is not connected to $\mathbf{z}_a$ and $\mathbf{z}_b$, we further assume that there exists another subset of observed variables $\mathbf{x}_a$ that is connected to $\mathbf{z}_a$ but not $\mathbf{z}_b$. Therefore, we have $\mathbf{z}_b \perp\!\!\!\perp \mathbf{x}_a | \mathbf{u}$. Similar to the derivation process in Theorem 1, suppose that we have different values of observed variables $\mathbf{x}_a$, by further conducting second-order partial derivatives w.r.t. $z_l, l \in [n]$ based on Equation (4), we can achieve component-wise identifiability by reducing the solution space of the full-rank linear system.

Intuitively, Theory 2 tells us about the degree to which each latent variable can be identified in the case of partial identifiability. For example, if we only have $M$ number of auxiliary variables, then when we try to determine a subset of latent variables $\mathbf{z}_a$ corresponding to two selected subsets of observed variables $\mathbf{x}_k$ and $\mathbf{x}_a$, if $M \geq 2|\mathbf{z}_a| + 1$ (we let $|\mathbf{z}_a|$ be the dimension of $\mathbf{z}_a$), then each $z_i \in \mathbf{z}_a$ is component-wise identifiable; otherwise, they are subspace identifiable. Moreover, if for any $\mathbf{x}_k$ and $\mathbf{x}_a$, the corresponding $\mathbf{z}_a$ with the largest dimension satisfies $M \geq 2|\mathbf{z}_a| + 1$, then the number of auxiliary variables is sufficient to achieve component-wise identifiability for all the latent variables. Since $|\mathbf{z}_a| \leq n$, we can use fewer values of the auxiliary variables to achieve component-wise identifiability. Please refer to Appendix D for further discussion of the theories.

Compared with existing works for component-wise identification Kong et al. (2022), our result is also more general. First, our method does not require any assumptions about the distribution of latent variables, while Khemakhem et al. (2020a); Hyvarinen et al. (2019); Hyvarinen & Morioka (2016) assume that the latent variables follow the exponential family distributions. Second, we exploit the conditional independence brought by the sparse mixing procedure to constrain the solution space of the linear full-rank system, so fewer values of auxiliary variables are required.

## 3.3 SPECIAL CASE WITH FULLY-CONNECTED MIXING PROCEDURE

When the mixing procedure is fully connected, we show that the existing identification results with auxiliary variables are special cases of our theory, as shown in Corollary 1.

**Corollary 1.** *(**General Case for Component-wise Identification with Full-connected Mixing procedure**) We follow the data generation process in Equation (1) and make assumptions A1, A2, and A3. In addition, we make the following assumptions:*

- *A5 (**Fully Connected Mixing procedure**): Each latent variable $z_i$ is connected to each observed variable $x_j$, where $i, j \in [n]$.*

*Suppose that we learn $\hat{g}$ to achieve Equation (1), $z_i$ is component-wise identifiable with $2n + 1$ different values of auxiliary variables $\mathbf{u}$.*

**Intuition of the Theoretical Results.** Please refer to Appendix A.3 for a proof. We provide the intuition behind the theoretical results as follows. Because the mixing procedure is fully connected,

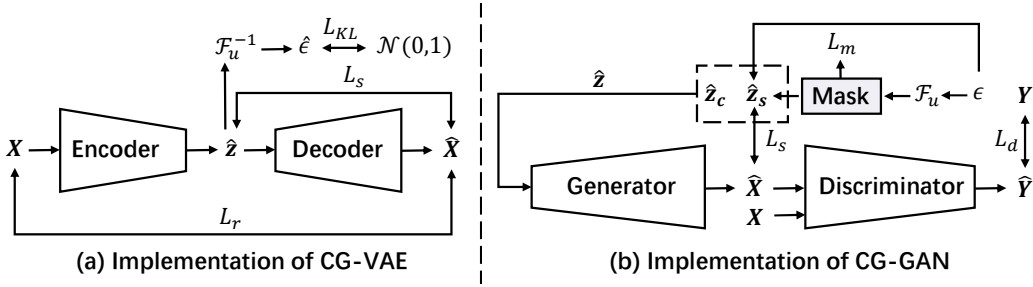

Figure 2: Illustrations of CG-VAE and CG-GAN, respectively. $\hat{\epsilon}$ and $\epsilon$ are the estimated and ground-truth noise variables, respectively. $\mathcal{F}_u$ denotes the domain-encoding neural networks. Note that in the CG-GAN, we partition the latent variables into the domain-invariant content variables $\mathbf{z}_c$ and domain-specific style $\mathbf{z}_s$. The $L_m$ denotes the mask restriction for automatically optimal dimension determination. And $Y$ and $\hat{Y}$ denote the ground-truth and predicted labels of real and fake samples.

there are no pairs of latent variables and observed variables that are independent conditioned on $\mathbf{u}$, so we cannot constrain the solution space of the linear equations obtained by the second-order partial derivatives. As a result, for $n$-dimension latent variables, $2n + 1$ auxiliary variables are needed to obtain compoenent-wise identifiability.

# 4 GENERAL GENERATIVE MODEL FRAMEWORK FOR DISENTANGLED REPRESENTATION LEARNING

To levarage the developed theory for disentanglement, we propose a generative model framework for disentangled representation learning, which includes the domain-encoding neural networks and the sparse mixing constraint. Additionally, we provide two practical implementations based on variational autoencoder (VAE) (Kingma, 2013) and generative adversarial network (GAN) (Goodfellow et al., 2020), respectively. For clarity, we name these implementations based on complementary gains as CG-VAE and CG-GAN, respectively. Moreover, We employ CG-VAE and CG-GAN for synthetic experiments and multi-domain image generation, respectively. Moreover, for a fair comparison, we follow the backbone architecture of (Xie et al., 2023), which also leverage nonlinear ICA and minimal change assumption to achieve identifiability of disentangled representation. The model architecture of the CG-VAE and CG-GAN are shown in Figure 2(a) and (b), respectively.

## 4.1 IMPLEMENTATION OF CG-VAE

We begin with the derivation of the evidence lower bound (ELBO) in Equation (6):

$$\log p(\mathbf{x}|\mathbf{u}) = \int \log p(\mathbf{x}|\mathbf{z})p(\mathbf{z}|\mathbf{u})d\mathbf{z} = \int \log \frac{p(\mathbf{x}|\mathbf{z})p(\mathbf{z}|\mathbf{u})q(\mathbf{z}|\mathbf{x})}{q(\mathbf{z}|\mathbf{x})}d\mathbf{z}$$

$$\geq \mathbb{E}_{q(\mathbf{z}|\mathbf{x})}\log p(\mathbf{x}|\mathbf{z}) + \mathbb{E}_{q(\mathbf{z}|\mathbf{x})}\log p(\mathbf{u}|\mathbf{z}) - KL(q(\mathbf{z}|\mathbf{x})||p(\mathbf{z}|\mathbf{u})), \tag{6}$$

where $KL$ denotes the Kullback–Leibler divergence. Since the reconstruction of $\mathbf{u}$ is not the optimization goal, we remove the reconstruction of $\mathbf{u}$. $q(\mathbf{z}|\mathbf{x})$ implemented as an encoder neural architecture, which outputs the mean and variance of the posterior distribution and $p(\mathbf{x}|\mathbf{z})$ is parameterized as the decoder that takes latent variables $\mathbf{z}$ for reconstruct observed variables.

**Domain-encoding Neural network.** Similar to Kong et al. (2022); Zhang et al. (2024), to model how auxiliary variables $\mathbf{u}$ influence the latent variables $\mathbf{z}$, we employ the normalizing flow-based architecture (Dinh et al., 2016; Huang et al., 2018; Durkan et al., 2019) that takes the estimated latent variables $\hat{\mathbf{z}}$ and auxiliary variables to turn it into $\hat{\epsilon}$. Specifically, we have:

$$\hat{\epsilon}, \log \det = \mathcal{F}_u(\hat{\mathbf{z}}), \tag{7}$$

where $\mathcal{F}_u$ is the normalizing flow and $\log \det$ is the log determinant of the conditional flow transformation on $\hat{\mathbf{z}}$. Therefore, by assuming that the noise term $\hat{\epsilon}$ follows a standard isotropic Gaussian, we can use the change of variable formula to compute the prior distribution of the latent variables:

$$\log p(\hat{\mathbf{z}}|\mathbf{u}) = \log p(\hat{\epsilon}) + \log \det. \tag{8}$$

**Sparse Mixing Constraint.** To enforce the conditional independence between the estimated latent variables and the observed variables, we further devise the sparse mixing constraint on the partial derivative of $\hat{x}$ with respect to $\hat{z}$, which are shown as follows:

$$L_s = \sum_{i,j \in [n]} \left| \frac{\partial \hat{x}_i}{\partial \hat{z}_j} \right|. \tag{9}$$

Finally, the total loss of the VAE-based implementation is shown as follows:

$$L_{vae} = \mathbb{E}_{q(\mathbf{z}|\mathbf{x})} \log p(\mathbf{x}|\mathbf{z}) + \alpha L_s - \beta KL(q(\mathbf{z}|\mathbf{x})||p(\mathbf{z}|\mathbf{u})) = L_r + \alpha L_s - \beta L_{KL}, \tag{10}$$

where $\alpha$ and $\beta$ are the hyper-parameters.

## 4.2 IMPLEMENTATION OF CG-GAN

We further provide the implementation of CG-GAN as shown in Figure 2 (b) since it is suitable for the multi-domain image translation task. In this case, we consider a more practical data generation process in Xie et al. (2023), where the latent variables are partitioned into the domain-invariant variable $\mathbf{z}_c$ and the domain-specific variables $\mathbf{z}_s$ for content and style, respectively.

**Domain-encoding neural networks** Similar to CG-VAE, we also use the domain-specific flow function $\mathcal{F}_u$ to embed the domain information. Following Xie et al. (2023), we use a learnable mask $\mathcal{M}$ as shown in the gray block in Figure 2 (b) to adaptively determine the dimension and location of $\mathbf{z}_s$ within the latent variable $\mathbf{z}$.

$$\hat{\mathbf{z}} = \boldsymbol{\epsilon} + \mathcal{M} \odot \mathcal{F}_u(\boldsymbol{\epsilon}) \tag{11}$$

where mask $\mathcal{M}$ is a vector with the same shape as $\boldsymbol{\epsilon}$. An entry of 0 in $\mathcal{M}$ indicates that the corresponding component belongs to $\mathbf{z}_c$ and will not be affected by domain changes. If the entry is not 0, it corresponds to $\mathbf{z}_s$, which will vary according to domain changes.

**Sparse Mixing Constraint** For enforcing the sparsity of the mixing procedure, we penalize the $l_1$-norm of the Jacobian vector of x with respect to each $z_s$, which is similar to our VAE-based architecture. Specifically, we use forward finite differences (Kim & Hong, 2021) to approximate the Jacobian vector in real-world experiments because the large dimensions of the image tensor make calculating the Jacobian computationally expensive.

$$L_s = \left\| \sum_{i=1}^{n_s} \frac{G(z + \epsilon_i) - G(z)}{\|\epsilon_i\|} \right\|_1$$

Here, $\epsilon_i$ is the perturbation vector, with only the $i$-th entry being non-zero. Finally, the total GAN-based model loss are shown as follows:

$$L_{gan} = L_d + \gamma L_m + \alpha L_s \tag{12}$$

where $\gamma$ and $\alpha$ are hyperparameters. $L_d$ is the traditional discriminator loss, $L_m$ is $l_1$-norm of the mask matrix, which is introduced by Xie et al. (2023), and the $L_s$ is the sparse mixing constraint.

## 5 EXPERIMENTS

### 5.1 SYNTHETIC EXPERIMENTS

#### 5.1.1 EXPERIMENT SETUP

**Data Generation.** We generate the synthetic data with multiple distributions. Specifically, we devise three datasets (Dataset A, B, and C) with 8, 9, and 11 numbers of domains, respectively. Please refer to Figure 6 for causal graphs of these data generation processes. The noise variables $\epsilon$ are sampled from a factorized Gaussian distribution for all datasets. We let the data generation process from latent variables to observed variables be MLPs with the LeaklyReLU activation function. Moreover, the dataset is randomly split into $90\%$ for training and $10\%$ for testing.

**Evaluation Metrics.** For evaluation, We use the mean correlation coefficient (MCC) (Hyvarinen & Morioka, 2016) between the true latent variables $\mathbf{z}$ and the estimated ones $\hat{\mathbf{z}}$. A higher MCC denotes the better identification performance the model can achieve. We also consider other metrics for

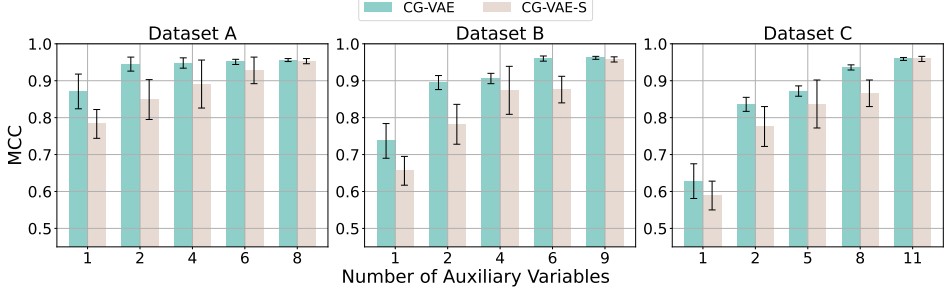

Figure 3: Experiments results on synthetic datasets. The horizontal axis represents the number of auxiliary variables, and the vertical axis represents the values of MCC.

Table 2: Mean correlation coefficient results on dataset A of different methods.

| Number of Domain | CG-VAE | CG-GAN | iMSDA | iVAE | FactorVAE | Slow-VAE | $\beta$-VAE |
|---|---|---|---|---|---|---|---|
| 2 | 0.953 (0.007) | 0.935 (0.016) | 0.869 (0.002) | 0.756 (0.105) | 0.728 (0.038) | 0.856 (0.053) | 0.810 (0.056) |
| 4 | 0.952 (0.017) | 0.927 (0.005) | 0.881 (0.043) | 0.807 (0.060) | 0.804 (0.031) | 0.798 (0.054) | 0.706 (0.083) |
| 6 | 0.956 (0.016) | 0.955 (0.007) | 0.936 (0.025) | 0.815 (0.029) | 0.755 (0.006) | 0.913 (0.031) | 0.795 (0.041) |
| 8 | 0.971 (0.009) | 0.944 (0.010) | 0.959 (0.017) | 0.920 (0.003) | 0.805 (0.004) | 0.921 (0.026) | 0.881 (0.045) |

disentanglement (Eastwood & Williams, 2018) like Completeness, Disentangle Score, Informativeness, $R^2$, and MSE. Appendix F provides the introduction of these metrics. Note that the higher the values of completeness, disentangle score, and R2, the better the performance of disentanglement, and the lower the values of informativeness, the better the performance of disentanglement. We repeat each experiment over 3 random seeds for each experiment and report the mean and standard deviation. Please refer to Appendix B for the implementation of the synthetic experiments.

**Baselines:** To evaluate the effectiveness of our method, we consider the following baselines. We first consider the standard $\beta$-VAE (Higgins et al., 2017), FactorVAE (Kim & Mnih, 2018), and SlowVAE (Klindt et al., 2020). We further consider other nonlinear ICA-based methods like iVAE (Khemakhem et al., 2020a) and iMSDA (Kong et al., 2022), which use auxiliary variables.

### 5.1.2 RESULTS AND DISCUSSION

The experimental results of the synthetic datasets are shown in Figure 3. To evaluate the effectiveness of the proposed sparsity constraint, we remove $L_s$ from CG-VAE and name it CG-VAE-S. According to the experiment results, we can obtain the following conclusions: **1)** when the number of auxiliary variables is sufficient, both CG-VAE and CG-VAE-S achieve relatively high MCC (around 0.95), demonstrating that sufficient changes brought by auxiliary variables benefit disentangled representation learning. **2)** as the number of auxiliary variables decreases, CG-VAE-S shows a noticeable decline in performance across the three datasets, while CG-VAE remains stable. In particular, for Dataset A, CG-VAE performs well even with just two domains, emphasizing the importance of sparse mixing constraints for achieving identifiable disentangled representation. **3)** When the number of domains is reduced to just one, both CG-VAE and CG-VAE-S perform poorly, indicating that sufficient variation is essential for disentangled representation. At the same time, we find that even with only a single domain, CG-VAE still performs better than CG-VAE-S, this is because the mapping from the ground truth to estimated latent variables is constrained by sparse mixing procedure, making certain latent variable subspaces identifiable. We also compare the proposed method with different baselines to evaluate the performance of disentanglement. Experiment results with the MCC, completeness, and disentanglement (Eastwood & Williams, 2018) are shown in Tables 2,3, and 4, respectively. Compared with the baselines, we can find that the proposed method achieves the best disentanglement performance in different metrics even the the domain number is limited, which verifies our theoretical results. Please refer to Appendix E for more experiment results.

Table 5: Results of two domain image generation on CelebA and MNIST datasets.

| Dataset | Metrics | TGAN | StyleGAN2-ADA | i-StyleGAN | CG-VAE | CG-GAN-M | CG-GAN |
|---|---|---|---|---|---|---|---|
| CelebA | FID ↓ | 4.89 | 3.57 | 2.65 | 3.02 | 2.60 | **2.57** |
|  | DIPD ↓ | 1.11 | 1.00 | 0.95 | **0.93** | **0.93** | **0.93** |
| MNIST | FID ↓ | 67.45 | 117.64 | 16.6 | 18.18 | 31.74 | **8.74** |
|  | Joint-FID ↓ | 155.21 | 386.19 | 107.39 | 111.76 | 81.53 | **67.04** |

Table 3: Completeness results on dataset A of different methods.

| Number of Domain | CG-VAE | CG-GAN | iMSDA | iVAE | FactorVAE | Slow-VAE | $\beta$-VAE |
|---|---|---|---|---|---|---|---|
| 2 | 0.608 (0.115) | 0.533 (0.074) | 0.405 (0.063) | 0.348 (0.035) | 0.349 (0.057) | 0.343 (0.072) | 0.319 (0.008) |
| 4 | 0.632 (0.054) | 0.659 (0.132) | 0.477 (0.028) | 0.358 (0.148) | 0.453 (0.069) | 0.356 (0.083) | 0.427 (0.095) |
| 6 | 0.626 (0.138) | 0.681 (0.062) | 0.663 (0.066) | 0.368 (0.029) | 0.529 (0.064) | 0.596 (0.107) | 0.525 (0.008) |
| 8 | 0.729 (0.064) | 0.809 (0.017) | 0.758 (0.042) | 0.435 (0.008) | 0.506 (0.057) | 0.687 (0.039) | 0.644 (0.113) |

Table 4: Disentanglement score results on dataset A of different methods.

| Number of Domain | CG-VAE | CG-GAN | iMSDA | iVAE | FactorVAE | Slow-VAE | $\beta$-VAE |
|---|---|---|---|---|---|---|---|
| 2 | 0.586 (0.118) | 0.529 (0.066) | 0.306 (0.032) | 0.347 (0.089) | 0.320 (0.091) | 0.347 (0.068) | 0.320 (0.009) |
| 4 | 0.589 (0.076) | 0.666 (0.127) | 0.498 (0.033) | 0.315 (0.002) | 0.380 (0.061) | 0.338 (0.080) | 0.403 (0.094) |
| 6 | 0.611 (0.125) | 0.672 (0.048) | 0.514 (0.045) | 0.548 (0.042) | 0.537 (0.078) | 0.620 (0.089) | 0.522 (0.036) |
| 8 | 0.697 (0.019) | 0.719 (0.013) | 0.710 (0.026) | 0.564 (0.076) | 0.626 (0.011) | 0.655 (0.073) | 0.692 (0.084) |

## 5.2 REAL-WORLD EXPERIMENTS

### 5.2.1 EXPERIMENT SETUP

**Datasets.** We use the CelebA ((Liu et al., 2015)) and the MNIST dataset ((LeCun et al., 1998)) for multi-domain image generation. To further evaluate the effectiveness of our theoretical results in real-world applications, we choose two domains for all the datasets. Specifically, in CelebA, we create two domains based on the presence or absence of eyeglasses, subsampling the no-eyeglasses domain to balance the sample sizes between the two. For MNIST, we use the training portion of the dataset and generate two domains consisting of red and green digits.

**Evaluation Metrics.** We evaluate our method using the Frechet Inception Distance (FID), a widely used metric for measuring the distribution divergence between generated and real images, where lower FID scores indicate better performance. Since the CelebA datasets lack paired data, we use Domain-Invariant Perceptual Distance (DIPD) to assess semantic correspondence (Liu et al., 2019). DIPD calculates the distance between instance-normalized Conv5 features of the VGG network. As for the MNIST dataset with ground truth tuples, we first compute the inception features of images in each tuple, and concatenate the features. Sequentially, we compute the Frechet distance between the features of the ground truth and generated tuples, referring to this as Joint-FID, as it measures the divergence between joint distributions. More addition, we remove the restriction of mask $\mathcal{M}$ and name this model variant as CG-GAN-M.

**Implementation and Baselines.** For a fair comparison, we build our method based on the official PyTorch implementation of i-StyleGAN (Xie et al., 2023), which also follows the backbone networks of StyleGAN2-ADA (Karras et al., 2020). Moreover, to verify the effectiveness of our introduced modules, we employ the default hyper-parameters of i-StyleGAN and only change the values of $\alpha$. As for the normalizing flow-based domain-encoding neural networks, we employ the deep sigmoid flow (DSF), which is designed to be component-wise strictly increasing. Since our method is built on StyleGAN2-ADA (Karras et al., 2020) and i-StyleGAN (Xie et al., 2023), we consider TGAN (Shahbazi et al. (2022)) as the compared method to evaluate the effectiveness of the sparse mixing constraint. Moreover, we further consider the CG-VAE, which uses the same decoder architecture as the generator of the i-StyleGAN. In practice, we train the Stylegan2-ADA and i-Stylegan for 25000k images, which is the default setting of Stylegan. For our method, to save the training time, we load the checkpoint of i-Stylegan at 20000k images, and train for 5000k images.

### 5.2.2 RESULTS AND ANALYSIS

**Comparison with Baselines.** Experimental results on the CelebaA and MNIST datasets with only two domains are shown in Table 5. We also provide some generated samples in Figure 4. According to the experiment results, we can find that the proposed method outperforms all other baselines on all the image generation tasks even though there are only two domains. Specifically, our method outperforms StyleGAN2-ADA with a clear margin, which reflects the importance of sufficient changes and the spare mixing restriction for disentangled representation learning. Moreover, compared with i-StyleGAN, we can find that our method also achieves a better result. We also find that the performance of CG-VAE is the worst, this is because the VAE-based models usually employ the Gaussian prior assumption (Bredell et al., 2023). The qualitative results shown in Figure 5 are clear. For example, in the adding glasses image generation task, we find that the images from i-StyleGAN change

TGAN       Stylegan2-ADA       i-Stylegan       CG-VAE       CG-GAN

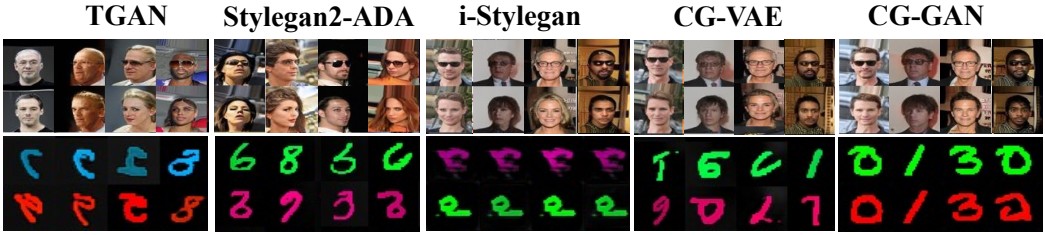

Figure 4: Samples of multi-domain image generation on the CelebA and MNIST datasets. i-StyleGAN, CG-VAE, and CG-GAN share the same noise input $\epsilon$. We find that when the number domain is insufficient, iStyleGAN will produce unnecessary changes.

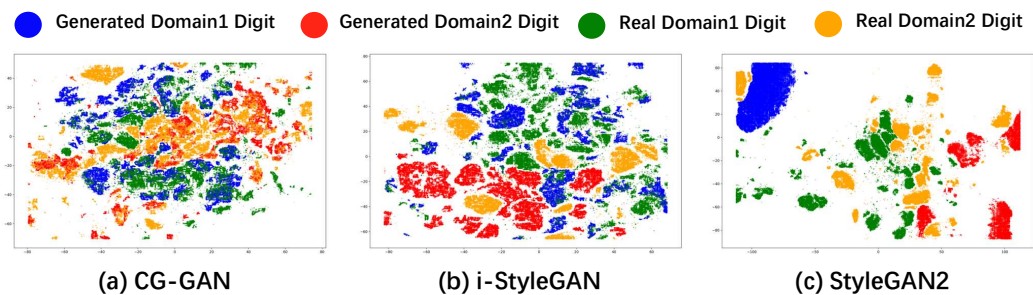

(a) CG-GAN                 (b) i-StyleGAN                 (c) StyleGAN2

Figure 5: The t-SNE visualization of different methods. Blue and red points represent features from generated images, while green and yellow points correspond to features from real images. A greater overlap between generated and real points within the same domain reflects improved performance.

the glasses and gender simultaneously. Meanwhile, the image generated from our method is stable, showing that our method can achieve better disentanglement under weak assumptions. Finally, we can find that the performance of CG-GAN-M is lower than that of standard CG-GAN, showing the necessity of the learnable mask for image generation. Similar to the several results of the VAE-based methods (Higgins et al., 2017; Pandey et al., 2021), the images generated by CG-VAE are less clear than those generated by CG-GAN due to their Gaussianity assumption.

**Visualization.** We also provide the t-SNE visualization, as shown in Figure 5. According to the experimental results, within the same domain, the data generated by our method shows the highest overlap with the corresponding ground truth.

## 6   CONCLUSION

This paper introduces a disentangled representation learning framework with identifiability guarantees by harnessing the complementary nature of sufficient changes and sparse mixing procedures, boosting its applicability to real-world scenarios. Specifically, the conditional independence induced by the sparse mixing procedure simplifies the mapping from estimated to ground truth latent variables, ensuring subspace identifiability. Meanwhile, sufficient changes promote componentwise identifiability of latent variables. Theoretical findings are validated through both synthetic and real-world experiments. Future work aims to extend these results to related tasks, such as transfer learning and causal representation learning. The empirical study in the paper preliminarily focuses on visual disentanglement–applications to more complex real-world scenarios are to be given.

## 7   ACKNOWLEDGMENT

We would like to acknowledge the support from NSF Award No. 2229881, AI Institute for Societal Decision Making (AI-SDM), the National Institutes of Health (NIH) under Contract R01HL159805, and grants from Quris AI, Florin Court Capital, and MBZUAI-WIS Joint Program. Moreover, this research was supported in part by the National Science and Technology Major Project (2021ZD0111501), Natural Science Foundation of China (U24A20233).

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

*Supplement to*
# "Synergy Between Sufficient Changes and Sparse Mixing Procedure for Disentangled Representation Learning"

Appendix organization:

# A PROOF

## A.1 SUBSPACE IDENTIFICATION VIA GENERALIZED SUFFICIENT CHANGES

**Theorem 1** (**Subspace Identification with Complementary Gains**). *Following the data generation process in Equation (1), we further make the following assumption.*

- *A1 (Smooth and Positive Density): The probability density function of latent variables is smooth and positive, i.e., $p_{\mathbf{z}|\mathbf{u}} > 0$ over $\mathcal{Z}$ and $\mathcal{U}$.*

- *A2 (Conditional Independence): Conditioned on $\mathbf{u}$, each $z_i$ is independent of any other $z_j$ for $i, j \in \{1, \cdots, n\}, i \neq j$, i.e., $\log p_{\mathbf{z}|\mathbf{u}}(\mathbf{z}|\mathbf{u}) = \sum_{i=1}^{n} \log p_{z_i|\mathbf{u}}(z_i|\mathbf{u})$.*

- *A3 (Generalized Sufficient Changes for Subspace Identification) Let $\mathbf{x}_k$ be a subset of $\mathbf{x}$ and $\mathbf{x}_{k,(1)}, \mathbf{x}_{k,(0)}$ be two different instance of $\mathbf{x}_k$. And vectors $w(k, \mathbf{u}) - w(k, 0)$ with $\mathbf{u} = 1, \ldots, |Pa(\mathbf{x}_k)|$ are linearly independent, where vector $w(k, \mathbf{u})$ is defined as:*

$$w(k, \mathbf{u}) = (\frac{\partial \log p(\mathbf{x}_{k,(1)}, \mathbf{x}_{\backslash k}|\mathbf{u})}{\partial z_1}, -\frac{\partial \log p(\mathbf{x}_{k,(0)}, \mathbf{x}_{\backslash k}|\mathbf{u})}{\partial z_1}, \cdots,$$
$$\frac{\partial \log p(\mathbf{x}_{k,(1)}, \mathbf{x}_{\backslash k}|\mathbf{u})}{\partial z_{Pa(\mathbf{x}_k)}}, -\frac{\partial \log p(\mathbf{x}_{k,(0)}, \mathbf{x}_{\backslash k}|\mathbf{u})}{\partial z_{Pa(\mathbf{x}_k)}}), \tag{13}$$

*Suppose that we learn $\hat{g}$ to achieve Equation (1) with the minimal number of edges of the mixing process. Then, for every pair of $\hat{z}_j$ and $\mathbf{x}_k$ that are not adjacent in the mixing process, we have $\mathbf{z}_{Pa(\mathbf{x}_k)}$ is not the function of $\hat{z}_j$, i.e., $\frac{\partial \mathbf{z}_{Pa(\mathbf{x}_k)}}{\partial \hat{z}_j} = 0$, where $\mathbf{z}_{Pa(\mathbf{x}_k)}$ is the parents of $\mathbf{x}_k$*

*Proof.* We start from the matched marginal distribution condition to develop the relation between $\mathbf{z}$ and $\hat{\mathbf{z}}$ as follows: $\forall \mathbf{u} \in \mathcal{U}$

$$p_{\hat{\mathbf{x}}|\mathbf{u}} = p_{\mathbf{x}|\mathbf{u}} \Longleftrightarrow p_{\hat{g}(\hat{\mathbf{z}})|\mathbf{u}} = p_{g(z)|\mathbf{u}} \Longleftrightarrow p_{g^{-1}\circ\hat{g}(\hat{g})|\mathbf{u}}|\mathbf{J}_{g^{-1}}| = p_{\mathbf{z}|\mathbf{u}}|\mathbf{J}|$$
$$\Longleftrightarrow p_{h(\hat{\mathbf{z}})|\mathbf{u}} = p_{\mathbf{z}|\mathbf{u}} \Longleftrightarrow \log p_{\hat{\mathbf{z}}|\mathbf{u}} - \log |\mathbf{J}| = \log p(\mathbf{z}|\mathbf{u}), \tag{14}$$

where $\hat{g} : \mathcal{Z} \rightarrow \mathcal{Z}$ denotes the estimated invertible generating function, and $h := g^{-1} \circ \hat{g}$ is the transformation between the true latent variable and the estimated one. $|\mathbf{J}_{g^{-1}}|$ stands for the absolute value of Jacobian matrix determinant of $g^{-1}$. Note that as both $\hat{g}^{-1}$ and $g$ are invertible, $|\mathbf{J}_{g^{-1}}| \neq 0$ and $h$ is invertible.

By matching the marginal distribution between $\hat{\mathbf{x}}$ and $\mathbf{x}$, we further take the first order derivative with $\hat{z}_j$ and have:

$$\frac{\partial \log p(\mathbf{x}|\mathbf{u})}{\partial \hat{z}_j} = \sum_{i=1}^{n} \frac{\partial \log p(\mathbf{x}|\mathbf{u})}{\partial z_i} \cdot \frac{\partial z_i}{\partial \hat{z}_j}. \tag{15}$$

Suppose there exists a group of observations $\mathbf{x}_k$ that are independent of $\hat{z}_j$ given $\mathbf{u}$, so there exists different values of $\mathbf{x}_k$, i.e., $\mathbf{x}_{k,(1)}$ and $\mathbf{x}_{k,(0)}$, making $\frac{\partial \log p(\mathbf{x}_k, \mathbf{x}_{\backslash k}|\mathbf{u})}{\partial \hat{z}_j}$ does not vary with different values of $\mathbf{x}_k$. Sequentially, we further let the other observed variables be $\mathbf{x}_{\backslash k}$ Then we subtract Equation (15) corresponding to $\mathbf{x}_{k,(1)}$ with that corresponding to $\mathbf{x}_{k,(0)}$ and have:

$$0 = \sum_{i=1}^{n} \Big( \frac{\partial \log p(\mathbf{x}_{k,(1)}, \mathbf{x}_{\backslash k}|\mathbf{u})}{\partial z_i} \cdot \frac{\partial z_i}{\partial \hat{z}_j}\Big|_{\mathbf{x}_k=\mathbf{x}_{k,(1)}} - \frac{\partial \log p(\mathbf{x}_{k,(0)}, \mathbf{x}_{\backslash k}|\mathbf{u})}{\partial z_i} \cdot \frac{\partial z_i}{\partial \hat{z}_j}\Big|_{\mathbf{x}_k=\mathbf{x}_{k,(0)}} \Big)$$
$$= \sum_{o \notin Pa(\mathbf{x}_k)} \Big( \frac{\partial \log p(\mathbf{x}_{k,(1)}, \mathbf{x}_{\backslash k}|\mathbf{u})}{\partial z_i} \cdot \frac{\partial z_i}{\partial \hat{z}_j}\Big|_{\mathbf{x}_k=\mathbf{x}_{k,(1)}} - \frac{\partial \log p(\mathbf{x}_{k,(0)}, \mathbf{x}_{\backslash k}|\mathbf{u})}{\partial z_i} \cdot \frac{\partial z_i}{\partial \hat{z}_j}\Big|_{\mathbf{x}_k=\mathbf{x}_{k,(0)}} \Big)$$
$$+ \sum_{i \in Pa(\mathbf{x}_k)} \Big( \frac{\partial \log p(\mathbf{x}_{k,(1)}, \mathbf{x}_{\backslash k}|\mathbf{u})}{\partial z_i} \cdot \frac{\partial z_i}{\partial \hat{z}_j}\Big|_{\mathbf{x}_k=\mathbf{x}_{k,(1)}} - \frac{\partial \log p(\mathbf{x}_{k,(0)}, \mathbf{x}_{\backslash k}|\mathbf{u})}{\partial z_i} \cdot \frac{\partial z_i}{\partial \hat{z}_j}\Big|_{\mathbf{x}_k=\mathbf{x}_{k,(0)}} \Big)$$
$$= \sum_{i \in Pa(\mathbf{x}_k)} \Big( \frac{\partial \log p(\mathbf{x}_{k,(1)}, \mathbf{x}_{\backslash k}|\mathbf{u})}{\partial z_i} \cdot \frac{\partial z_i}{\partial \hat{z}_j}\Big|_{\mathbf{x}_k=\mathbf{x}_{k,(1)}} - \frac{\partial \log p(\mathbf{x}_{k,(0)}, \mathbf{x}_{\backslash k}|\mathbf{u})}{\partial z_i} \cdot \frac{\partial z_i}{\partial \hat{z}_j}\Big|_{\mathbf{x}_k=\mathbf{x}_{k,(0)}} \Big), \tag{16}$$

where $\text{Pa}(\mathbf{x}_k)$ denotes the indices set of the parents of $\mathbf{x}_k$. Similarly, we further have:

$$0 = \sum_{i \in \text{Pa}(\mathbf{x}_k)} \left( \frac{\partial \log p(\mathbf{x}_{k,(2)}, \mathbf{x}_{\setminus k}|\mathbf{u})}{\partial z_i} \cdot \frac{\partial z_i}{\partial \hat{z}_j} \bigg|_{\mathbf{x}_k=\mathbf{x}_{k,(2)}} - \frac{\partial \log p(\mathbf{x}_{k,(0)}, \mathbf{x}_{\setminus k}|\mathbf{u})}{\partial z_i} \cdot \frac{\partial z_i}{\partial \hat{z}_j} \bigg|_{\mathbf{x}_k=\mathbf{x}_{k,(0)}} \right)$$

$$0 = \sum_{i \in \text{Pa}(\mathbf{x}_k)} \left( \frac{\partial \log p(\mathbf{x}_{k,(2)}, \mathbf{x}_{\setminus k}|\mathbf{u})}{\partial z_i} \cdot \frac{\partial z_i}{\partial \hat{z}_j} \bigg|_{\mathbf{x}_k=\mathbf{x}_{k,(2)}} - \frac{\partial \log p(\mathbf{x}_{k,(1)}, \mathbf{x}_{\setminus k}|\mathbf{u})}{\partial z_i} \cdot \frac{\partial z_i}{\partial \hat{z}_j} \bigg|_{\mathbf{x}_k=\mathbf{x}_{k,(1)}} \right)$$

$$(17)$$

Suppose that we let $\mathbf{u} = \mathbf{u}_0, \cdots, \mathbf{u}_{|\text{Pa}(x_k)|}$, then by combining Equation (16) and (17) and further leveraging the sufficient changes assumption (A3), the linear system is a $3|\text{Pa}(x_k)| \times 3|\text{Pa}(x_k)|$ full-rank system. Therefore, the only solution is $\frac{\partial z_i}{\partial \hat{z}_j}\big|_{\mathbf{x}_k=\mathbf{x}_{k,(0)}} = 0, \frac{\partial z_i}{\partial \hat{z}_j}\big|_{\mathbf{x}_k=\mathbf{x}_{k,(1)}} = 0$, and $\frac{\partial z_i}{\partial \hat{z}_j}\big|_{\mathbf{x}_k=\mathbf{x}_{k,(2)}} = 0, i \in \text{Pa}(\mathbf{x}_k)$, implying that $\frac{\partial z_i}{\partial \hat{z}_j} = 0, i \in \text{Pa}(\mathbf{x}_k)$.

As $h(\cdot)$ is smooth $\mathcal{Z}$, its Jacobian can written as

$$\boldsymbol{J}_h = \left[ \begin{array}{c|c} \frac{\partial \mathbf{z}_o}{\partial \hat{\mathbf{z}}_o} \neq 0 & \frac{\partial \mathbf{z}_o}{\partial \hat{\mathbf{z}}_s} \\ \hline \frac{\partial \mathbf{z}_s}{\partial \hat{\mathbf{z}}_o} = 0 & \frac{\partial \mathbf{z}_s}{\partial \hat{\mathbf{z}}_s} \end{array} \right]. \tag{18}$$

Therefore, $\frac{\partial \mathbf{z}_o}{\partial \hat{\mathbf{z}}_o} \neq 0$ and $\mathbf{z}_o$ is subspace identifiable. □

## A.2 COMPONENT-WISE IDENTIFICATION VIA SUFFICIENT CHANGES OF MULTIPLE DISTRIBUTIONS

**Theorem 2.** *(Component-wise Identification with Complementary Gains) Let the observations be sampled from the data generating process in Equation (1). Suppose that Assumptions from Theorem 1 hold and also that we learn $\hat{g}, p_{\hat{\mathbf{z}}|\mathbf{u}}$ to match the marginal distribution between $\mathbf{x}$ and $\hat{\mathbf{x}}$ with the minimal number of edges of the mixing process. We further make the following assumption:*

- *A4 (Sufficient Changes for Component-wise Identification.): Suppose $\mathbf{x}_k$ and $\mathbf{x}_a$ be a subset of $\mathbf{x}$ and $\mathbf{x}_k \cap \mathbf{x}_a = \emptyset$. We let the $\mathbf{z}_a$ be the set of latent variables that connect to $\mathbf{x}_a$ but not connect to $\mathbf{x}_k$, such that for all different values of $\mathbf{x}_a$ such as $\mathbf{x}_{a,0}, \mathbf{x}_{a,(1)}$, the vectors $\mathcal{V}(a, k, \mathbf{u}_s) - \mathcal{V}(a, k, 0)$ with $s = 1, \ldots, 2|\mathbf{z}_a|$ are linearly independent. And the vector $\mathcal{V}(a, k, \mathbf{u}_s)$ is defined as follows:*

$$\mathcal{V}(a, k, \mathbf{u}_m) = \Big( \frac{\partial^2 \log p(\mathbf{z}_{a,(1)}, \mathbf{z}_b, \mathbf{z}_k|\mathbf{u}_m)}{(\partial z_i)^2}, -\frac{\partial^2 \log p(\mathbf{z}_{a,(0)}, \mathbf{z}_b, \mathbf{z}_k|\mathbf{u}_m)}{(\partial z_i)^2}, \ldots,$$
$$\frac{\partial \log p(\mathbf{z}_{a,(1)}, \mathbf{z}_b, \mathbf{z}_k|\mathbf{u}_m)}{\partial z_i}, -\frac{\partial \log p(\mathbf{z}_{a,(0)}, \mathbf{z}_b, \mathbf{z}_k|\mathbf{u}_m)}{\partial z_i} \Big)_{z_i \in \mathbf{z}_a}. \tag{19}$$

*Suppose that we learn $\hat{g}$ to achieve Equation (1) with the minimal number of edges of the mixing process. Then $\mathbf{z}_a$ is component-wise identifiable.*

*Proof.* Based on Theorem 1, given different values of $\mathbf{z}_k$ and $|\text{Pa}(\mathbf{x}_k)|$ different domains, Equation (15) can be further derived as follows:

$$\frac{\partial \log p(\mathbf{x}|\mathbf{u})}{\partial \hat{z}_j} = \sum_{i=1}^{n} \frac{\partial \log p(\mathbf{x}|\mathbf{u})}{\partial z_i} \cdot \frac{\partial z_i}{\partial \hat{z}_j} = \sum_{z_i \notin \mathbf{z}_k} \frac{\partial \log p(\mathbf{x}|\mathbf{u})}{\partial z_i} \cdot \frac{\partial z_i}{\partial \hat{z}_j}. \tag{20}$$

Sequentially, if there exists $\mathbf{x}_a, \mathbf{x}_a \cap \mathbf{x}_k = \emptyset$, that connects to $\hat{z}_j$ and $\mathbf{z}_a$ but does not connect to $\mathbf{z}_b$. Similar to Theorem 1, given two different values of $\mathbf{x}_{a,(1)}$ and $\mathbf{x}_{a,(0)}$, we subtract Equation (20) that

corresponding to $\mathbf{x}_{a,(1)}$ with that corresponding to $\mathbf{x}_{a,(0)}$ and then we have:

$$
\frac{\partial \log p(\mathbf{x}_{a,(1)}, \mathbf{x}_b, \mathbf{z}_k|\mathbf{u})}{\partial \hat{z}_j} - \frac{\partial \log p(\mathbf{x}_{a,(0)}, \mathbf{x}_b, \mathbf{z}_k|\mathbf{u})}{\partial \hat{z}_j}
$$
$$
= \sum_{\mathbf{z}_i \notin \mathbf{z}_k} \left( \frac{\partial \log p(\mathbf{x}_{a,(1)}, \mathbf{x}_b, \mathbf{z}_k|\mathbf{u})}{\partial z_i} \cdot \frac{\partial z_i}{\partial \hat{z}_j}\Big|_{\mathbf{x}_a=\mathbf{x}_{a,(1)}} - \frac{\partial \log p(\mathbf{x}_{a,(0)}, \mathbf{x}_b, \mathbf{z}_k|\mathbf{u})}{\partial z_i} \cdot \frac{\partial z_i}{\partial \hat{z}_j}\Big|_{\mathbf{x}_a=\mathbf{x}_{a,(0)}} \right)
$$
$$
= \sum_{z_i \notin \mathbf{z}_k \text{and} z_i \in \mathbf{z}_a} \left( \frac{\partial \log p(\mathbf{x}_{a,(1)}, \mathbf{x}_b, \mathbf{z}_k|\mathbf{u})}{\partial z_i} \cdot \frac{\partial z_i}{\partial \hat{z}_j}\Big|_{\mathbf{x}_a=\mathbf{x}_{a,(1)}} - \frac{\partial \log p(\mathbf{x}_{a,(0)}, \mathbf{x}_b, \mathbf{z}_k|\mathbf{u})}{\partial z_i} \cdot \frac{\partial z_i}{\partial \hat{z}_j}\Big|_{\mathbf{x}_a=\mathbf{x}_{a,(0)}} \right)
$$
$$(21)$$

By combining Equation (14) and (21), we have:

$$
\frac{\partial \log p(\hat{\mathbf{z}}_{a,(1)}, \hat{\mathbf{z}}_b, \hat{\mathbf{z}}_k|\mathbf{u})}{\partial \hat{z}_j} - \frac{\partial \log p(\hat{\mathbf{z}}_{a,(0)}, \hat{\mathbf{z}}_b, \hat{\mathbf{z}}_k|\mathbf{u})}{\partial \hat{z}_j} - \frac{\partial \log |\boldsymbol{J}_h|}{\partial \hat{z}_j}\Big|_{\mathbf{x}_a=\mathbf{x}_{a,(1)}} + \frac{\partial \log |\boldsymbol{J}_h|}{\partial \hat{z}_j}\Big|_{\mathbf{x}_a=\mathbf{x}_{a,(0)}}
$$
$$
= \sum_{z_i \notin \mathbf{z}_k} \left( \frac{\partial \log p(\hat{\mathbf{z}}_{a,(1)}, \hat{\mathbf{z}}_b, \hat{\mathbf{z}}_k|\mathbf{u})}{\partial z_i} \cdot \frac{\partial z_i}{\partial \hat{z}_j}\Big|_{\mathbf{x}_a=\mathbf{x}_{a,(1)}} - \frac{\partial \log p(\hat{\mathbf{z}}_{a,(0)}, \hat{\mathbf{z}}_b, \hat{\mathbf{z}}_k|\mathbf{u})}{\partial z_i} \cdot \frac{\partial z_i}{\partial \hat{z}_j}\Big|_{\mathbf{x}_a=\mathbf{x}_{a,(0)}} \right)
$$
$$
= \sum_{z_i \notin \{\mathbf{z}_k\} \text{and} z_i \in \mathbf{z}_a} \left( \frac{\partial \log p(\hat{\mathbf{z}}_{a,(1)}, \hat{\mathbf{z}}_b, \hat{\mathbf{z}}_k|\mathbf{u})}{\partial z_i} \cdot \frac{\partial z_i}{\partial \hat{z}_j}\Big|_{\mathbf{x}_a=\mathbf{x}_{a,(1)}} - \frac{\partial \log p(\hat{\mathbf{z}}_{a,(0)}, \hat{\mathbf{z}}_b, \hat{\mathbf{z}}_k|\mathbf{u})}{\partial z_i} \cdot \frac{\partial z_i}{\partial \hat{z}_j}\Big|_{\mathbf{x}_a=\mathbf{x}_{a,(0)}} \right)
$$
$$
= \sum_{z_i \in \mathbf{z}_a} \left( \frac{\partial \log p(\hat{\mathbf{z}}_{a,(1)}, \hat{\mathbf{z}}_b, \hat{\mathbf{z}}_k|\mathbf{u})}{\partial z_i} \cdot \frac{\partial z_i}{\partial \hat{z}_j}\Big|_{\mathbf{x}_a=\mathbf{x}_{a,(1)}} - \frac{\partial \log p(\hat{\mathbf{z}}_{a,(0)}, \hat{\mathbf{z}}_b, \hat{\mathbf{z}}_k|\mathbf{u})}{\partial z_i} \cdot \frac{\partial z_i}{\partial \hat{z}_j}\Big|_{\mathbf{x}_a=\mathbf{x}_{a,(0)}} \right),
$$
$$(22)$$

where $\boldsymbol{J}_{h_0}$ and $\boldsymbol{J}_{h_1}$ denote the the Jacobian with the values of $\mathbf{x}_{a,(1)}$ and $\mathbf{x}_{a,(0)}$, respectively.

And then we further take the second-order derivative w.r.t $\hat{z}_l$, where $l \neq j$ and have:

$$
0 - \frac{\partial \log |\boldsymbol{J}_h|}{\partial \hat{z}_j \partial \hat{z}_l}\Big|_{\mathbf{x}_a=\mathbf{x}_{a,(1)}} + \frac{\partial \log |\boldsymbol{J}_h|}{\partial \hat{z}_j \partial \hat{z}_l}\Big|_{\mathbf{x}_a=\mathbf{x}_{a,(0)}}
$$
$$
= \sum_{z_i \in \mathbf{z}_a} \left( \frac{\partial^2 \log p(\mathbf{z}|\mathbf{u})}{(\partial z_i)^2} \cdot \frac{\partial z_i}{\partial \hat{z}_j} \cdot \frac{\partial z_i}{\partial \hat{z}_l}\Big|_{\mathbf{x}_a=\mathbf{x}_{a,(1)}} - \frac{\partial^2 \log p(\mathbf{z}|\mathbf{u})}{(\partial z_i)^2} \cdot \frac{\partial z_i}{\partial \hat{z}_j} \cdot \frac{\partial z_i}{\partial \hat{z}_l}\Big|_{\mathbf{x}_a=\mathbf{x}_{a,(0)}} \right) +
$$
$$
+ \sum_{z_i \in \mathbf{z}_a} \left( \frac{\partial \log p(\mathbf{z}|\mathbf{u})}{\partial z_i} \cdot \frac{\partial^2 z_i}{\partial \hat{z}_j \partial \hat{z}_l}\Big|_{\mathbf{x}_a=\mathbf{x}_{a,(1)}} - \frac{\partial \log p(\mathbf{z}|\mathbf{u})}{\partial z_i} \cdot \frac{\partial^2 z_i}{\partial \hat{z}_j \partial \hat{z}_l}\Big|_{\mathbf{x}_a=\mathbf{x}_{a,(0)}} \right)
$$
$$
= \sum_{z_i \in \mathbf{z}_a} \left( v(a,k,i,\mathbf{u}) \cdot \frac{\partial z_i}{\partial \hat{z}_j} \cdot \frac{\partial z_i}{\partial \hat{z}_l}\Big|_{\mathbf{x}_a=\mathbf{x}_{a,(1)}} - v(a,k,i,\mathbf{u}) \cdot \frac{\partial z_i}{\partial \hat{z}_j} \cdot \frac{\partial z_i}{\partial \hat{z}_l}\Big|_{\mathbf{x}_a=\mathbf{x}_{a,(0)}} \right.
$$
$$
\left. + \overline{v}(a,k,i,\mathbf{u}) \cdot \frac{\partial^2 z_i}{\partial \hat{z}_j \partial \hat{z}_l}\Big|_{\mathbf{x}_a=\mathbf{x}_{a,(1)}} - \overline{v}(a,k,i,\mathbf{u}) \cdot \frac{\partial^2 z_i}{\partial \hat{z}_j \partial \hat{z}_l}\Big|_{\mathbf{x}_a=\mathbf{x}_{a,(0)}} \right)
$$
$$(23)$$

Then we consider $\mathbf{u} = \mathbf{u}_0, \cdots, \mathbf{u}_{|\mathbf{z}_a|}$, and let Equation (23) corresponding to $\mathbf{u}_s$ subtract with that corresponding to $\mathbf{u}_0$, and have:

$$
0 = \sum_{z_i \in \mathbf{z}_a} \left( v(a,k,i,\mathbf{u}_s) \cdot \frac{\partial z_i}{\partial \hat{z}_j} \cdot \frac{\partial z_i}{\partial \hat{z}_l}\Big|_{\mathbf{x}_a=\mathbf{x}_{a,(1)}} - v(a,k,i,\mathbf{u}_s) \cdot \frac{\partial z_i}{\partial \hat{z}_j} \cdot \frac{\partial z_i}{\partial \hat{z}_l}\Big|_{\mathbf{x}_a=\mathbf{x}_{a,(0)}} \right.
$$
$$
+ \overline{v}(a,k,i,\mathbf{u}_s) \cdot \frac{\partial^2 z_i}{\partial \hat{z}_j \partial \hat{z}_l}\Big|_{\mathbf{x}_a=\mathbf{x}_{a,(1)}} - \overline{v}(a,k,i,\mathbf{u}_s) \cdot \frac{\partial^2 z_i}{\partial \hat{z}_j \partial \hat{z}_l}\Big|_{\mathbf{x}_a=\mathbf{x}_{a,(0)}} \Bigg) -
$$
$$
\sum_{z_i \in \mathbf{z}_a} \left( v(a,k,i,\mathbf{u}_0) \cdot \frac{\partial z_i}{\partial \hat{z}_j} \cdot \frac{\partial z_i}{\partial \hat{z}_l}\Big|_{\mathbf{x}_a=\mathbf{x}_{a,(1)}} - v(a,k,i,\mathbf{u}_0) \cdot \frac{\partial z_i}{\partial \hat{z}_j} \cdot \frac{\partial z_i}{\partial \hat{z}_l}\Big|_{\mathbf{x}_a=\mathbf{x}_{a,(0)}} \right.
$$
$$
+ \overline{v}(a,k,i,\mathbf{u}_0) \cdot \frac{\partial^2 z_i}{\partial \hat{z}_j \partial \hat{z}_l}\Big|_{\mathbf{x}_a=\mathbf{x}_{a,(1)}} - \overline{v}(a,k,i,\mathbf{u}_0) \cdot \frac{\partial^2 z_i}{\partial \hat{z}_j \partial \hat{z}_l}\Big|_{\mathbf{x}_a=\mathbf{x}_{a,(0)}} \Bigg)
$$
$$(24)$$

Under the generalized sufficient changes assumptions and further let three different values of $\mathbf{x}_a$, i.e., $\mathbf{x}_{a,(0)}, \mathbf{x}_{a,(1)}$ and $\mathbf{x}_{a,(2)}$, so the linear system is a $6|\mathbf{z}_a| \times 6|\mathbf{z}_a|$ full-rank system. Therefore, the only solution is $\frac{\partial z_i}{\partial \hat{z}_j} \cdot \frac{\partial z_i}{\partial \hat{z}_l} = 0$ and $\frac{\partial^2 z_i}{\partial \hat{z}_j \partial \hat{z}_l} = 0$.

Note that $\frac{\partial z_i}{\partial \hat{z}_j} \cdot \frac{\partial z_i}{\partial \hat{z}_l} = 0$ implies that for each $z_i \in \mathbf{z}_a$, $\frac{\partial z_i}{\partial \hat{z}_j} \neq 0$ for at most one element $k \in [n]$. Therefore, there is only at most one non-zero entry in each row indexed by $z_i \in \mathbf{z}_a$ in the Jacobian $\boldsymbol{J}_h$. As a result, $\mathbf{z}_a$ is component-wise identifiable with $n_{k,a}$ different domains, where $n_{k,a} = \max(|\text{Pa}(\mathbf{x}_k)|, 2 \times |\mathbf{z}_a| + 1)$. Therefore, for each latent variable $z_i, i \in [n]$, the necessary domain number for the component-wise identifiability is $\max(n_{k,a})$. □

### A.3 GENERAL CASE FOR COMPONENT-WISE IDENTIFICATION WITH FULL-CONNECTED MIXING PROCESS

**Corollary 1.** *(**General Case for Component-wise Identification with Full-connected Process**) We follow the data generation process in Equation (1) and make assumptions A1, A2, and A3. In addition, we make the following assumptions:*

- *A6 (**Fully Connected Mixing Process**): Each latent variable $z_i$ is connected to each observed variable $x_j$, where $i, j \in [n]$.*

*Suppose that we learn $\hat{g}$ to achieve Equation (1), $z_i$ is component-wise identifiable with $2n + 1$ different values of auxiliary variables $\mathbf{u}$.*

*Proof.* We conduct second-order derivation of Equation (14) w.r.t $\hat{z}_j$ and $\hat{z}_l$ and have:

$$0 = \frac{\partial \log p(\hat{\mathbf{z}}|\mathbf{u})}{\partial \hat{z}_j \partial \hat{z}_l} = \sum_{i=1}^{n} \left( \frac{\partial^2 \log p(z_i|\mathbf{u})}{(\partial z_i)^2} \cdot \frac{\partial z_i}{\partial \hat{z}_j} \cdot \frac{\partial z_i}{\partial \hat{z}_l} + \frac{\partial \log p(z_i|\mathbf{u})}{\partial z_i} \cdot \frac{(\partial z_i)^2}{\partial \hat{z}_j \partial \hat{z}_l} \right) + \frac{\partial \log |\boldsymbol{J}_h|}{\partial \hat{z}_j \partial \hat{z}_l}. \quad (25)$$

Suppose we have $2n + 1$ different values of $\mathbf{u}$, i.e., $\mathbf{u} = \mathbf{u}_0, \mathbf{u}_1, \cdots, \mathbf{u}_{2n}$, let Equation (25 corresponding to $\mathbf{u}_s$ subtract with that corresponding to $\mathbf{u}_0$, and have:

$$0 = \sum_{i=1}^{n} \left( \left( \frac{\partial^2 \log p(z_i|\mathbf{u}_s)}{(\partial z_i)^2} - \frac{\partial^2 \log p(z_i|\mathbf{u}_0)}{(\partial z_i)^2} \right) \cdot \frac{\partial z_i}{\partial \hat{z}_j} \cdot \frac{\partial z_i}{\partial \hat{z}_l} + \left( \frac{\partial \log p(z_i|\mathbf{u}_s)}{\partial z_i} - \frac{\partial \log p(z_i|\mathbf{u}_0)}{\partial z_i} \right) \cdot \frac{(\partial z_i)^2}{\partial \hat{z}_j \partial \hat{z}_l} \right). \quad (26)$$

With assumption A4, the linear system is $2n \times 2n$ full-rank system, and the unique solution is $\frac{\partial z_i}{\partial \hat{z}_j} \cdot \frac{\partial z_i}{\partial \hat{z}_l} = 0$ and $\frac{(\partial z_i)^2}{\partial \hat{z}_j \partial \hat{z}_l} = 0$. Since $\boldsymbol{J}_h$ is invertible and full-rank, for each row of $\boldsymbol{J}_h$, there is only one non-zero element, implying that latent variables $\mathbf{z}$ are component-wise identifiable. □

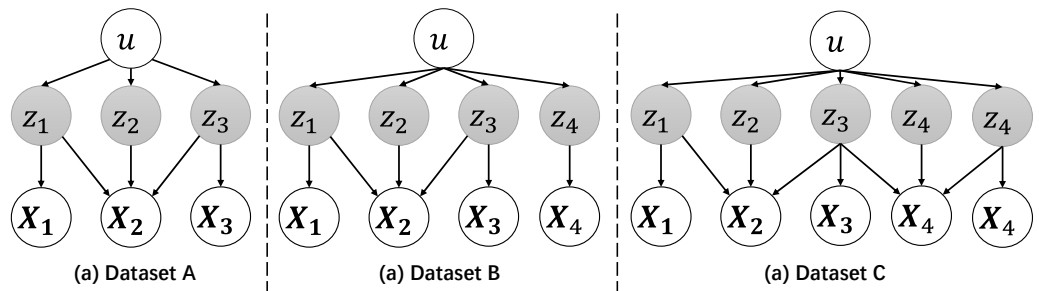

Figure 6: Data generation process of the three datasets for synthetic data.

# B SYNTHETIC EXPERIMENTS

## B.1 SIMULATION DATASETS

The synthetic datasets are constructed in accordance with the causal graphs as shown in Figure 6. Specifically, the mixing functions are synthesized using multilayer perceptrons (MLPs) initialized with random weights. The mixing functions incorporate the LeakyReLU activation function to ensure invertibility. For each domain, we set different noises to ensure that **z** has different distributions.

## B.2 MEAN CORRELATION COEFFICIENT

Mean Correlation Coefficient, a widely used metric in ICA research, evaluates how well latent factors are recovered. It works by calculating the absolute correlation coefficients between each ground truth factor and the estimated latent variables. If the recovered factors involve component-wise invertible nonlinearities, either Pearson's or Spearman's rank correlation coefficients are used accordingly. The best matching between ground truth and estimated factors is found by solving a linear sum assignment problem on the correlation matrix, which is efficiently solvable in polynomial time.

## B.3 IMPLEMENTATION DETAILS

Both the encoder and decoder in the proposed CG-VAE are composed of a 5-layer multilayer perception (MLP) with the LeakyReLU activate function. 2. We trained the VAE network using the AdamW optimizer with a learning rate of $3e-3$ and a mini-batch size of 64, the s. The coefficients for $\alpha$ and $\beta$ are $5e-2$ and $1e-3$, respectively.

# C RELATED WORKS

## C.1 DISENTANGLED REPRESENTATION LEARNING

Disentangled representation learning plays an important role in unsupervised learning (Li et al.; Locatello et al., 2020b; Peters et al., 2017). The key intuition (Bengio et al., 2013) is that the disentangled representations should separate the distinct, independent, and informative generative factors of variation in the data. And each latent variable is sensitive to changes in single underlying generative factors while being relatively invariant to changes in other factors. Based on this intuition, researchers further extend this concept to groups of factors (Bouchacourt et al., 2018; Suter et al., 2018; Cai et al., 2019; Higgins et al.). Several methods (Wang et al., 2022) are proposed to learn disentangled representations. For example, some researchers employ the variational autoencoders Kingma (2013); Burgess et al. (2018); Kumar et al. (2017); Kim & Mnih (2018) to learn disentangled representation. Other works employ the GAN-based structures to learn disentangled representations. For instance, InfoGAN Chen et al. (2016) achieves disentanglement by leveraging an extra variational regularization of mutual information. Recent works (Locatello et al., 2019a; 2020a;b) propose that unsupervised learning of disentangled representations is impossible, without any inductive biases on both the models and the data. Therefore, Locatello et al. (2020c) leverages pairs of observations to learn disentangled representation in a weakly-supervised. And Träuble et al.

(2021) learns disentangled representation from correlations data. Recently, Fumero et al. (2023) obtains disentangled representation in multi-task models by considering that the features activate sparsely with respect to tasks and are maximally shared across tasks.

## C.2 NONLINEAR INDEPENDENT COMPONENT ANALYSIS

Independent component analysis (ICA) provides the theoretical foundation for disentangled representation (Hyvärinen et al., 2023) and identifiability is an important challenge. Previously, several works achieve identifiability by assuming the mixing processes are linear (Comon, 1994; Hyvärinen & Pajunen, 1999; Hyvärinen, 2013). However, the nonlinear ICA is a challenging task since the latent variables are not identifiable without any extra assumptions on the distribution of latent variables or the generation process (Hyvärinen et al., 2024; Khemakhem et al., 2020b; Zheng et al., 2022). To solve this problem, some works leverage the auxiliary variables e.g. domain indexes, time indexes, and class labels to achieve sufficient changes for identification (Hyvarinen & Morioka, 2016; Khemakhem et al., 2020a; Kong et al., 2022; Li et al., 2023; Hälvä & Hyvarinen, 2020; Hälvä et al., 2021; Hyvarinen & Morioka, 2017). Recently, other works achieve identifiability by using the sparse causal model. Specifically, Lachapelle et al. (2022; 2024); Zhang et al. (2024); Li et al. (2024) employ sparse interactions between latent variables to achieve identifiability. And Moran et al. (2021); Zheng & Zhang (2023); Zheng et al. (2022) achieve identifiability by using structural sparsity assumption. However, either the sufficient changes or the structural sparsity assumptions are hard to meet in real-world scenarios. Fortunately, we find the complementary benefits between the sufficient changes and sparse mixing procedure, so we can mitigate these assumptions.

## C.3 MULTI-DOMAIN IMAGE GENERATION

Multi-domain image generation focuses on learning the joint distribution of multi-domain image data, even without paired data (Liu & Tuzel, 2016; Pu et al., 2018; Xie et al., 2023). CoGAN Liu & Tuzel (2016) uses different generators for each domain while sharing high-level weights to transfer domain-invariant information across domains. JointGAN Pu et al. (2018) factorizes the joint distribution into marginal and conditional distributions to learn. i-StyleGAN Xie et al. (2023) connects multi-domain generation with identifiability by dividing the latent input $z$ into $z_c$ and $z_s$, where $z_c$ captures domain-invariant information, and $z_s$ captures domain-specific information. Unlike conditional GANs (Gong et al., 2019; Brock, 2018; Miyato & Koyama, 2018; Odena et al., 2017; Kang & Park, 2020; Kang et al., 2021; Zhang et al., 2019), which aim to generate images that conform to provided label information and enhance diversity to approximate the marginal distribution of each domain, multi-domain image generation seeks to model the relationships between different domains within a unified framework. This approach not only captures the individual domain distributions but also the shared and domain-specific characteristics across domains. When transitioning between domains, the objective is to make minimal adjustments necessary to adapt to the new domain while preserving as much domain-invariant information as possible.

# D DISCUSSION OF THE THEORETICAL RESULTS

## D.1 IMPLICATIONS

Based on the aforementioned theoretical results, we can relax the assumptions required for disentangled representation learning, which holds practical significance in real-world scenarios. For example, in the task of multi-domain image generation, using previous methods, it is challenging to obtain the necessary number of domains to identify all latent variables. Although existing work introduces the assumption of minimal change to alleviate this issue to some extent, due to the unknown dimensions of the latent variables, the required number of domains still does not meet theoretical needs, resulting in unsatisfactory practical outcomes.

Meanwhile, our method has two advantages in solving such problems. First, the conditional independence caused by the sparse mixing procedure constrains the solution space of the linear full-rank system, thus decreasing the requirement on the auxiliary variables. Second, even with an insufficient number of values of the auxiliary variables, we can still disentangle the latent variables that are of interest and relevant to downstream tasks (e.g., $\mathbf{z}_{o_1}$ in Figure **??**), even if other unrelated latent

Table 6: Experiment results of different model variants of CG-VAE.

| Number of Domain | MCC | | | | Completeness | | | |
|---|---|---|---|---|---|---|---|---|
| | CG-VAE | CG-FactorVAE | CG-TCVAE | CG-HFVAE | CG-VAE | CG-FactorVAE | CG-TCVAE | CG-HFVAE |
| 2 | 0.953 | 0.882 | 0.946 | 0.944 | 0.608 | 0.814 | 0.553 | 0.633 |
| 4 | 0.952 | 0.876 | 0.900 | 0.932 | 0.632 | 0.732 | 0.661 | 0.497 |
| 6 | 0.956 | 0.837 | 0.918 | 0.907 | 0.620 | 0.626 | 0.600 | 0.612 |
| 8 | 0.971 | 0.869 | 0.934 | 0.956 | 0.729 | 0.590 | 0.793 | 0.719 |

variables are not identifiable. For instance, in multi-domain image generation (such as controlled image generation with or without glasses), it is sufficient to identify the latent variable related to a particular feature, such as glasses, with a small number of domains. Therefore, our method is more likely to meet practical needs.

### D.2 INTUITION OF SUBSPACE-WISE AND COMPONENT-WISE IDENTIFICATION

Subspace identification is weaker than component-wise identification, and hence naturally requires less information in the data generation process and distributions. Let us start with the subspace identifiability, suppose we have two sets of latent variables with changing distributions denoted by $\mathbf{z}_s$ and $\mathbf{z}_c$, respectively. And $\hat{\mathbf{z}}_s$ and $\hat{\mathbf{z}}_c$ are the corresponding estimated variables. Moreover, we let the $i$- and $j$- component of $\mathbf{z}_s$ and $\mathbf{z}_c$ be $z_s^i$ and $z_c^j$, respectively. In order to see whether $\mathbf{z}_s$ is identifiable up to a subspace, we just need to show whether $\frac{\partial z_s^i}{\partial \hat{z}_c^j} = 0$. To this end, only the first-order derivative of the logarithmic density distribution is needed. However, for the propose of component-wise identifiability, we have to make additional conditions, such as conditional independence between $z_s^i$ and $z_s^j$ given the all the remaining components. These conditions impose stronger constraints compared to subspace identifiability. Mathematically, this conditional independence implies that the second-order cross derivative of the logarithmic density distribution with respect to $\hat{z}_i$ and $\hat{z}_j$ is zero, which involves second-order derivatives.

### D.3 DISCUSSION OF ASSUMPTIONS OF THEOREM 2

In contrast to subspace identification, here we assume that the conditional distributions are second-order differentiable, and linear independence is necessary for a unique solution to the full-rank linear system. Additionally, while the conditional independence between $\mathbf{z}_{o_2}$ and $\mathbf{x}_r$ implies a sparse mixing procedure, our assumption on the sparsity is weaker than structural sparsity (Zheng et al., 2022; Zheng & Zhang, 2023). For example, the case shown in Figure 2 achieves component-wise identifiability but violates the structural sparsity assumption. How to further analyze the relationship between structural sparsity and the sparse procedure proposed in this paper is an interesting future direction. We also note that sparse mixing procedures are common in real-world tasks. In multi-domain image generation, for instance, when transforming an image of a man into a woman, the latent variable representing gender affects only certain observed features, such as facial hair and hairstyle, while leaving the background and other facial features unchanged.

## E   MORE EXPERIMENT RESULTS

### E.1   SCALABILITY OF DIFFERENT VAE VARIANTS

We also extend our method to other VAE variants like TC-VAE Chen et al. (2018), FactorVAE Kim & Mnih (2018), and HFVAE Esmaeili et al. (2018). We named them CG-TCVAE, CG-FactorVAE, and CG-HFVAE, respectively. experiment on synthetic are shown in Table 6. According to the experiment, we can find that the all the model variants achieve the ideal disentanglement performance even the number of domains is 2, which verify our theoretical results.

### E.2   OTHER CONSTRAINT ON MIXING PROCEDURE

In this paper, we choose $L_1$-norm to enforce sparsity of mixing procedure since the L1 penalty induces a "sharp" constraint, encouraging the penalized quantities (like the coefficients in linear regression or the partial derivatives in our formulation) to become exactly zero during optimization.

Table 7: MCC results of the CG-VAE and CG-VAE-L2.

| Number of Domains | CGVAE-L1 | CGVAE-L2 |
|---|---|---|
| 2 | 0.953 (0.007) | 0.887 (0.071) |
| 4 | 0.952 (0.017) | 0.924 (0.011) |
| 6 | 0.956 (0.016) | 0.892 (0.025) |
| 8 | 0.971 (0.009) | 0.911 (0.035) |

Table 8: Informativeness results on dataset A of different methods.

| Number of Domain | CG-VAE | CG-GAN | iMSDA | iVAE | FactorVAE | Slow-VAE | beta-VAE |
|---|---|---|---|---|---|---|---|
| 2 | 0.265 (0.018) | 0.291 (0.028) | 0.317 (0.028) | 0.402 (0.164) | 0.525 (0.013) | 0.350 (0.046) | 0.343 (0.045) |
| 4 | 0.256 (0.045) | 0.351 (0.032) | 0.361 (0.045) | 0.288 (0.007) | 0.532 (0.040) | 0.399 (0.014) | 0.428 (0.036) |
| 6 | 0.212 (0.016) | 0.259 (0.019) | 0.282 (0.001) | 0.251 (0.090) | 0.480 (0.017) | 0.288 (0.051) | 0.318 (0.037) |
| 8 | 0.206 (0.016) | 0.265 (0.038) | 0.225 (0.013) | 0.216 (0.018) | 0.471 (0.012) | 0.322 (0.037) | 0.278 (0.051) |

This characteristic is particularly desirable in scenarios where we aim to simplify the model or identify the most relevant components in the mixing process. In the meanwhile, the L2 norm (squared sum) tends to stabilize training by uniformly penalizing the magnitude of all coefficients, it does not promote sparsity as effectively. Instead, it typically results in coefficients with small but non-zero values, which may not align with our objective in this context. Therefore, the L1-norm can be used to induce sparsity of the estimated mixing procedure, which aligns with the goal of our theoretical results. We replaced L2-norm with L1-norm, and conducted experiments on the synthetic datasets, which are shown in Table 7. According to the experiment, we can find that disentanglement performance of the model with L2-norm lower than that of L1, evaluating our statement.

### E.3 EXPERIMENT RESULTS ON OTHER METRICS

Experiment results on the Dataset A of informativeness, $R^2$, and MSE are shown in Tables 8, 9, and 10, respectively.

## F METRICS INTRODUCTION

To quantitatively analyze the disentanglement performance of different methods and settings, we employ six metrics: MCC, Completeness, Disentanglement, Informativeness, MSE, and R-squared.

**MCC (Mean Correlation Coefficient)** evaluates the correspondence between the latent variables and generative factors. It calculates the correlation matrix between the latent representations and the generative factors and uses the Hungarian algorithm to maximize the absolute values of the diagonal elements after sorting. The final MCC score is the mean of the absolute values of the diagonal elements.

**Completeness (Eastwood & Williams, 2018):** This metric measures the degree to which each generative factor is captured by a single latent variable. It requires a training regressor to predict generative factors from latent variables and computes the entropy of the importance weights of latent variables for each factor.

**Disentanglement (Eastwood & Williams, 2018)**: This metric assesses whether each latent variable is primarily associated with a single generative factor. It is computed by training regressors to predict latent variables from generative factors and calculating the entropy of their importance distribution.

**Informativeness (Eastwood & Williams, 2018)**: his metric quantifies the total amount of information about the generative factors contained in the latent variables. It is computed as the root mean squared error (RMSE) between the predicted generative factors and the ground truth **z**.

**Mean Squared Error (MSE) (Li et al., 2023)**: By using LASSO as the regressor to predict **z** from the latent representation **ẑ**, we use MSE between he predicted generative factors and the ground truth values on the test set to evaluate the disentanglement performance.

$R^2$: The coefficient of determination, which is also a common metric for the regression task.

Table 9: $R^2$ results on dataset A of different methods.

| Number of Domain | CG-VAE | CG-GAN | iMSDA | iVAE | FactorVAE | Slow-VAE | $\beta$-VAE |
|---|---|---|---|---|---|---|---|
| 2 | 0.931 (0.002) | 0.916 (0.014) | 0.854 (0.019) | 0.843 (0.090) | 0.656 (0.012) | 0.854 (0.015) | 0.848 (0.003) |
| 4 | 0.930 (0.023) | 0.876 (0.014) | 0.883 (0.015) | 0.904 (0.008) | 0.728 (0.063) | 0.834 (0.014) | 0.799 (0.085) |
| 6 | 0.950 (0.006) | 0.937 (0.009) | 0.925 (0.002) | 0.940 (0.004) | 0.766 (0.016) | 0.900 (0.049) | 0.908 (0.013) |
| 8 | 0.955 (0.002) | 0.938 (0.034) | 0.946 (0.016) | 0.941 (0.006) | 0.790 (0.005) | 0.897 (0.038) | 0.912 (0.021) |

Table 10: MSE results on dataset A of different methods.

| Number of Domain | CG-VAE | CG-GAN | iMSDA | iVAE | FactorVAE | Slow-VAE | $\beta$-VAE |
|---|---|---|---|---|---|---|---|
| 2 | 0.072 (0.009) | 0.084 (0.017) | 0.105 (0.009) | 0.122 (0.020) | 0.325 (0.009) | 0.141 (0.030) | 0.158 (0.013) |
| 4 | 0.074 (0.031) | 0.154 (0.032) | 0.120 (0.005) | 0.064 (0.006) | 0.311 (0.037) | 0.158 (0.003) | 0.215 (0.010) |
| 6 | 0.055 (0.005) | 0.072 (0.015) | 0.080 (0.000) | 0.067 (0.009) | 0.233 (0.003) | 0.118 (0.044) | 0.137 (0.020) |
| 8 | 0.048 (0.005) | 0.086 (0.001) | 0.052 (0.007) | 0.052 (0.002) | 0.234 (0.010) | 0.118 (0.040) | 0.081 (0.028) |

