# OpenReview forum: "Synergy Between Sufficient Changes and Sparse Mixing Procedure for Disentangled Representation Learning"
_ICLR.cc/2025/Conference — ICLR 2025 Poster_

### Official Review · Reviewer_qELo · 2024-10-28

**Soundness:** 2
**Presentation:** 2
**Contribution:** 3
**Rating:** 6
**Confidence:** 4

**Summary:**

In this paper, authors combine two assumptions for disentangling latent variables (latent identifiability). Thew whole work is based on non-linear ICA (identifiable VAE) with auxiliary variables $u$. Authors show that sufficient changes on the distribution of latent variables and exploiting structural sparsity assumptions on the mixing procedure can be complement each other to achieve better identifiability. Theorems are illustrated and proved in this paper. With both synthetic and real-world experiments, authors show that there proposed approach based on VAE and GAN can achieve good latent indentifiability.

**Strengths:**

* The theories in this paper is presented and illustrated in a clear way. The schematic graphical model is good. Besides, there are solid prooves of those theorems.
* The contribution is good, I can visually understand the usefulness of the proposed method.
* There are both synthetic results validating the method, and real-world applications to show that the new method is useful.

**Weaknesses:**

* The sentence of Theorem A3 is a bit confusing. It is a bit hard to read that long sentence.
* Line 158, why $x_k$ contribute $z_j$? And the derivative $\frac{\partial z_j}{\partial x_k}$. Could the author explain why the relationship is not $z_j$ contribute $x_k$.
* Line 349, "$q(z|x)$  is the an encoder"
* Eq. 9, why loss $L_s$ is chosen to be the absolute sum rather than the squared sum? Will a squared sum be easier to train and more stable? Any reason or trade-off analysis on this choice?
Or does absolute sum have sparsity?
* What is CG-VAE-S? It seems like it was used before it was introduced.
* For synthetic results, why not apply CG-GAN? Also, there are no other baseline methods for comparison.
* It is also very common to compare the generated image with the true by MSE, $R^2$, or cross-entropy (log-likelihood). They are intuitive metrics that are worth seeing.
* Same for the real-world, why not try CG-VAE. If there is some simple reason, authors should state in one sentence why VAE is not suitable for real-world datasets. Also, are there any other alternative methods for real-world, since all four methods are actually some method combinations? Please correct me if my understanding is wrong.
* To me, it would be better to have more experimental results in the main content. The current experiment section is a bit simple. I'd like to see for example, more comprehensive comparison.

**Questions:**

/

---

> ### Author Response · Authors · 2024-11-22
> **Response to Reviewer qELo, Part 1**
>
> Dear qELo, thank you for taking the time and effort to review our paper. Your valuable suggestions have been instrumental in helping us improve the soundness of our experiments. We deeply appreciate your insightful feedback and thoughtful comments.
>
> >W1: The sentence of Theorem A3 is a bit confusing. It is a bit hard to read that long sentence.
>
> **A1**: Thank you for pointing this out. To address this, we have revised the statement by breaking it into two shorter sentences, which we believe will make it clearer and easier to follow. Specifically, the revised A3 is:
>
> **A3 (Generalized Sufficient Changes for Subspace Identification):** Let $\mathcal{x} _k$ be a subset of $\mathcal{x}$ and $\mathcal{x} _{k, (1)}, \mathcal{x} _{k, 0}$ be two different instance of $\mathcal{x} _k$. And vectors $w(k, \mathcal{u})-w(k,0)$ with $\mathcal{u}=1,\dots,|\text{Pa}(\mathcal{x} _k)|$ are linearly independent.
>
> We hope this modification resolves your concern, and we appreciate your feedback for improving the readability of our work.
>
>
>
> >W2: Line 158, why $x_k$ contribute $z _j$? And the derivative $\frac{\partial z _j}{\partial x _k}$. Could the author explain why the relationship is not $z_j$ contribute $x _k$?
>
> **A2**: Thanks for your question. We hope you will find the intuition in **Line 78-79**. Specifically, by using the sparsity constraint, we can model the ground truth mixing procedure. If $\hat{z}_j$ is not adjacent to $x _k$, i.e., $\hat{z} _j$ contribute $x _k$, then $\frac{\partial \mathcal{z} _{\text{Pa}(x _k)}}{\partial \hat{z} _j} \equiv 0$. In light of your question, we have highlighted this in Lines 169-170.
>
> >W3: Line 349, "$q(z|x)$ is the an encoder"
>
> **A3**: Thanks for your careful review. When we refer to $q(\mathcal{z}|\mathcal{x})$ as 'an encoder', we mean that $q(\mathcal{z}|\mathcal{x})$ is implemented as an encoder neural architecture in practice. Specifically, $q(\mathcal{z}|\mathcal{x})$ is used to approximate the posterior distribution over the latent variable $\mathcal{z}$ given observed variables $\mathcal{x}$. The encoder parameterizes the posterior distribution through a mean and variance in a probabilistic framework of variational inference [1]. Therefore, $q(\mathcal{z}|\mathcal{x})$ serves as the practical tool for inferring $\mathcal{z}$ from $\mathcal{x}$. In light of your suggestion, we have modified it in Line 322 of the revised version.
>
> [1] Kingma, Diederik P. "Auto-encoding variational bayes." arXiv preprint arXiv:1312.6114 (2013).
>
> >W4: Eq. 9, why loss $L_s$ is chosen to be the absolute sum rather than the squared sum? Will a squared sum be easier to train and more stable? Any reason or trade-off analysis on this choice? Or does absolute sum have sparsity?
>
> **A4**: Thank you for your insightful question. The choice of the absolute sum for $L_s$, corresponding to the L1 norm in statistics [2], was deliberate. Specifically, the L1 penalty induces a "sharp" constraint, encouraging the penalized quantities (like the coefficients in linear regression or the partial derivatives in our formulation) to become exactly zero during optimization. This characteristic is particularly desirable in scenarios where we aim to simplify the model or identify the most relevant components in the mixing process. In the meanwhile, the L2 norm (squared sum) tends to stabilize training by uniformly penalizing the magnitude of all coefficients, it does not promote sparsity as effectively. Instead, it typically results in coefficients with small but non-zero values, which may not align with our objective in this context. Therefore, the L1-norm can be used to induce sparsity of the estimated mixing procedure, which aligns with the goal of our theoretical results. In light of your questions, we have replaced L2-norm, i.e., the squared sum, with L1-norm, and conducted experiments on the synthetic datasets, which are shown in the following tables.
>
> | MCC ↑   | CG-VAE | CG-VAE-L2 |
> |-------------------|--------|-------|
> | Number of Domains |                    |                 |
> |         2         | 0.953 (0.007) | 0.887 (0.071) |
> |         4         | 0.952 (0.017) | 0.924 (0.011) |
> |         6         | 0.956 (0.016) | 0.892 (0.025) |
> |         8         | 0.971 (0.009) | 0.911 (0.035) |
>
> In light of your question, we have provided the discussion and experiment results in Appendix E.2.
>
> [2] Tibshirani, Robert. "Regression shrinkage and selection via the lasso." Journal of the Royal Statistical Society Series B: Statistical Methodology 58.1 (1996): 267-288.

---

> ### Author Response · Authors · 2024-11-22
> **Response to Reviewer qELo, Part 2**
>
> >W5: What is CG-VAE-S? It seems like it was used before it was introduced.
>
> **A5**: Thanks a lot! We apologize for the oversight in the original version of the paper. CG-VAE-S is a variant of the CG-VAE model, where the sparsity constraint is removed. We have now added a clear explanation of CG-VAE-S in the revised version of the paper to ensure that its introduction is properly aligned with its usage. In light of your question, we have added it in the Line 416 of the reversed version.
>
> >W6: For synthetic results, why not apply CG-GAN? Also, there are no other baseline methods for comparison.
>
> **A6**: Thank you for your valuable feedback. As per your suggestion, we have included CG-GAN in our synthetic experiments as shown in Lines 390-395 and Lines 432-444 of the revised paper. Additionally, we have considered a wider range of baseline methods for comparison, including various disentangled representation learning approaches such as IMSDA [3], iVAE [4], $\beta$-VAE [5], FactorVAE [6], and SloW-VAE [7]. Experiment results are shown as follows:
>
> | MCC ↑         | CG-VAE          | CG-GAN          | iMSDA      | iVAE         | beta-VAE     | FactorVAE    | Slow-VAE     |
> |--------------------|--------------|--------------|--------------|--------------|--------------|--------------|--------------|
> | Number of Domains   |                  |                  |                  |                  |                  |                  |                  |
> | 2                  | 0.953 (0.007)   | 0.935 (0.016)   | 0.869 (0.002)   | 0.756 (0.105)   | 0.810 (0.056)   | 0.728 (0.038)   | 0.856 (0.053)   |
> | 4                  | 0.952 (0.017)   | 0.927 (0.005)   | 0.881 (0.043)   | 0.807 (0.060)   | 0.706 (0.083)   | 0.804 (0.031)   | 0.798 (0.054)   |
> | 6                  | 0.956 (0.016)   | 0.955 (0.007)   | 0.936 (0.025)   | 0.815 (0.029)   | 0.795 (0.041)   | 0.755 (0.006)   | 0.913 (0.031)   |
> | 8                  | 0.971 (0.009)   | 0.944 (0.010)   | 0.959 (0.017)   | 0.920 (0.003)   | 0.881 (0.045)   | 0.805 (0.004)   | 0.921 (0.026)   |
>
>
>
> [3] Kong, Lingjing, et al. "Partial disentanglement for domain adaptation." International conference on machine learning. PMLR, 2022.
> [4] Khemakhem, Ilyes, et al. "Variational autoencoders and nonlinear ica: A unifying framework." International conference on artificial intelligence and statistics. PMLR, 2020.
> [5] Higgins, Irina, et al. "beta-vae: Learning basic visual concepts with a constrained variational framework." ICLR (Poster) 3 (2017).
> [6] Kim, Hyunjik, and Andriy Mnih. "Disentangling by factorising." International conference on machine learning. PMLR, 2018.
> [7]Klindt, David, et al. "Towards nonlinear disentanglement in natural data with temporal sparse coding." arXiv preprint arXiv:2007.10930 (2020).
>
> >W7: It is also very common to compare the generated image with the true by MSE, $R^2$, or cross-entropy (log-likelihood). They are intuitive metrics that are worth seeing.
>
> **A7**: Thank you for your thoughtful suggestion. Since metrics like MSE and $R^2$ rely on ground-truth latent variables for estimation, they are difficult to calculate in real-world tasks where these ground-truth variables are unknown. Therefore, we have included comparisons using MSE and $R^2$ as additional evaluation metrics on the synthetic experiments. Specifically, we computed MSE based on the approach outlined in [8], and $R^2$ following the method in [3]. Regarding your point about cross-entropy, since the latent variables in our model are continuous, cross-entropy is not an appropriate metric for evaluating disentanglement performance. Instead, we have adopted completeness, disentanglement score, and informativeness as disentanglement evaluation metrics [9]. Experiment results are shown as follows.
>
> | MSE ↓             | CG-VAE          | CG-GAN          | iMSDA      | iVAE         | beta-VAE     | FactorVAE    | Slow-VAE     |
> |-------------------|--------------|--------------|--------------|--------------|--------------|--------------|--------------|
> | Number of Domains  |              |              |              |              |              |              |              |
> | 2                 | 0.072 (0.009)| 0.084 (0.017)| 0.105 (0.009)| 0.122 (0.020)| 0.158 (0.013)| 0.325 (0.009)| 0.141 (0.030)|
> | 4                 | 0.074 (0.031)| 0.154 (0.032)| 0.120 (0.005)| 0.064 (0.006)| 0.215 (0.010)| 0.311 (0.037)| 0.158 (0.003)|
> | 6                 | 0.055 (0.005)| 0.072 (0.015)| 0.080 (0.000)| 0.067 (0.009)| 0.137 (0.020)| 0.233 (0.003)| 0.118 (0.044)|
> | 8                 | 0.048 (0.005)| 0.086 (0.001)| 0.052 (0.007)| 0.052 (0.002)| 0.081 (0.028)| 0.234 (0.010)| 0.118 (0.040)|

---

> ### Author Response · Authors · 2024-11-22
> **Response to Reviewer qELo, Part 3**
>
> | R-square ↑         | CG-VAE          | CG-GAN          | iMSDA      | iVAE         | beta-VAE     | FactorVAE    | Slow-VAE     |
> |--------------------|--------------|--------------|--------------|--------------|--------------|--------------|--------------|
> | Number of Domains   |              |              |              |              |              |              |              |
> | 2                  | 0.931 (0.002) | 0.916 (0.014) | 0.854 (0.019) | 0.843 (0.090) | 0.848 (0.003) | 0.656 (0.012) | 0.854 (0.015) |
> | 4                  | 0.930 (0.023) | 0.876 (0.014) | 0.883 (0.015) | 0.904 (0.008) | 0.799 (0.085) | 0.728 (0.063) | 0.834 (0.014) |
> | 6                  | 0.950 (0.006) | 0.937 (0.009) | 0.925 (0.002) | 0.940 (0.004) | 0.908 (0.013) | 0.766 (0.016) | 0.900 (0.049) |
> | 8                  | 0.955 (0.002) | 0.938 (0.034) | 0.946 (0.016) | 0.941 (0.006) | 0.912 (0.021) | 0.790 (0.005) | 0.897 (0.038) |
>
>
>
> | Completeness ↑         | CG-VAE          | CG-GAN          | iMSDA      | iVAE         | beta-VAE     | FactorVAE    | Slow-VAE     |
> |--------------------|--------------|--------------|--------------|--------------|--------------|--------------|--------------|
> | Number of Domains   |              |              |              |
> | 2                       | 0.608 (0.115)| 0.533 (0.074)| 0.405 (0.063)| 0.348 (0.035)| 0.319 (0.008)| 0.349 (0.057)| 0.343 (0.072)|
> | 4                       | 0.632 (0.054)| 0.659 (0.132)| 0.477 (0.028)| 0.358 (0.148)| 0.427 (0.095)| 0.453 (0.069)| 0.356 (0.083)|
> | 6                       | 0.626 (0.138)| 0.681 (0.062)| 0.663 (0.066)| 0.368 (0.029)| 0.525 (0.008)| 0.529 (0.064)| 0.596 (0.107)|
> | 8                       | 0.729 (0.064)| 0.809 (0.017)| 0.758 (0.042)| 0.435 (0.008)| 0.644 (0.113)| 0.506 (0.057)| 0.687 (0.039)|
>
> | Disentanglement Score ↑         | CG-VAE          | CG-GAN          | iMSDA      | iVAE         | beta-VAE     | FactorVAE    | Slow-VAE     |
> |--------------------|--------------|--------------|--------------|--------------|--------------|--------------|--------------|
> | Number of Domains   |              |              |              |
> | 2                  | 0.586 (0.118)    | 0.529 (0.066)    | 0.306 (0.032) | 0.347 (0.089) | 0.320 (0.009) | 0.320 (0.091) | 0.347 (0.068) |
> | 4                  | 0.589 (0.076)    | 0.666 (0.127)    | 0.498 (0.033) | 0.315 (0.002) | 0.403 (0.094) | 0.380 (0.061) | 0.338 (0.080) |
> | 6                  | 0.611 (0.125)    | 0.672 (0.048)    | 0.514 (0.045) | 0.548 (0.042) | 0.522 (0.036) | 0.537 (0.078) | 0.620 (0.089) |
> | 8                  | 0.697 (0.019)    | 0.719 (0.013)    | 0.710 (0.026) | 0.564 (0.076) | 0.692 (0.084) | 0.626 (0.011) | 0.655 (0.073) |
>
> | Informativeness ↓         | CG-VAE          | CG-GAN          | iMSDA      | iVAE         | beta-VAE     | FactorVAE    | Slow-VAE     |
> |--------------------|--------------|--------------|--------------|--------------|--------------|--------------|--------------|
> | Number of Domains   |              |              |              |
> | 2                  | 0.265 (0.018)   | 0.291 (0.028)   | 0.317 (0.028) | 0.402 (0.164) | 0.343 (0.045) | 0.525 (0.013) | 0.350 (0.046) |
> | 4                  | 0.256 (0.045)   | 0.351 (0.032)   | 0.361 (0.045) | 0.288 (0.007) | 0.428 (0.036) | 0.532 (0.040) | 0.399 (0.014) |
> | 6                  | 0.212 (0.016)   | 0.259 (0.019)   | 0.282 (0.001) | 0.251 (0.090) | 0.318 (0.037) | 0.480 (0.017) | 0.288 (0.051) |
> | 8                  | 0.206 (0.016)   | 0.265 (0.038)   | 0.225 (0.013) | 0.216 (0.018) | 0.278 (0.051) | 0.471 (0.012) | 0.322 (0.037) |
>
> Thanks for your reminder, we have added this experiment results in Appendix E.3.
>
> [8] Li, Zijian, et al. "Subspace identification for multi-source domain adaptation." Advances in Neural Information Processing Systems 36 (2024).
> [9] Eastwood, Cian, and Christopher KI Williams. "A framework for the quantitative evaluation of disentangled representations." 6th International Conference on Learning Representations. 2018.

---

> ### Author Response · Authors · 2024-11-22
> **Response to Reviewer qELo, Part 4**
>
> >W8: Same for the real-world, why not try CG-VAE. If there is some simple reason, authors should state in one sentence why VAE is not suitable for real-world datasets. Also, are there any other alternative methods for real-world, since all four methods are actually some method combinations? Please correct me if my understanding is wrong.
>
> **A8**: Thank you for your valuable comments. We have included CG-VAE in our experiments with real-world datasets, and the results are shown in the table below. Please refer to the visualization results in Figure 5 of the revised version.
>
> | Dataset  | Metrics | TGAN  | StyleGAN2-ADA | i-StyleGAN | CG-GAN-M | CG-GAN | CG-VAE |
> |----------|---------|-------|---------------|------------|----------|--------|--------|
> | CelebA   | FID ↓   | 4.89  | 3.57          | 2.65       | 2.60     | 2.57   | 3.02      |
> | MNIST    | FID ↓   | 67.45 | 117.64        | 16.6       | 31.74    | 8.74   | 18.18     |
>
> Regarding your question on VAE suitability: While VAEs are widely used, we found that their image generation performance on real-world datasets was suboptimal due to their tendency to produce blurry or less detailed images. This is because of the injected noise and imperfect reconstruction, and with the standard decoder (with factorized output distribution), the generated samples are much more blurred than those coming from GANs. There are other alternative methods like diffusion models, but they require significant computational resources, which makes them less practical for the scope of our current work.

---

> > ### Comment · Reviewer_qELo · 2024-11-22
> >
> > Thanks for these responses, I'd love to raise my score from 5 to 6. I hope the author can rewrite the results part of the paper to incorporate some of these important comparisons if it is accepted.

---

> > > ### Author Response · Authors · 2024-11-22
> > >
> > > We are very happy that you found the response well addressed your concerns. Thank you once again for your valuable comments and suggestions and for championing our submission. In light of your suggestions, we have incorporated these important comparisons and rewritten the experiment section.
> > >
> > > With best wishes,
> > >
> > > Authors of submission 4906

---

### Official Review · Reviewer_4KGP · 2024-11-03

**Soundness:** 2
**Presentation:** 2
**Contribution:** 2
**Rating:** 6
**Confidence:** 4

**Summary:**

In this paper, the authors investigated disentangled representation learning and proposed a new framework by combining two existing methods: sufficient changes and sparse mixing procedures. This framework can guarantee identifiability of the models, because the sparse mixing procedure can induce conditional independence between latent variables , which can simplify the mapping from estimated latent to ground truth. On the other hand, sufficient changes can further improve component-wise identifiability of latent variables. They implemented this framework for two types of deep generative models: VAEs and GANs. They evaluated on multiple datasets and show that their approach can result in better disentangled representations.

**Strengths:**

Quality and Clarity: In general this paper is in good quality. The paper is well structured and the motivation is significant, and the writing is clear. The authors provide very detailed math derivations for the framework, and to my best knowledge, the derivations are sound.

Originality: While the paper leverages two existing methods for disentangled representation learning, it is interesting that they explore a framework that combines them in a compatible way, and this framework will also guarantee the identifiability of representation learning.

**Weaknesses:**

While they validated their framework on multiple datasets, they only provided qualitative results in terms of the disentanglement of learned representations. I would hope that they can evaluated the results more quantitatively and make comparison based on metrics of disentanglement, e.g. the metrics used in [1] and [2]. I guess the reason I want to see this is it is difficult to see the disentanglement only based on t-SNE plots, because t-SNE itself will employ an algorithm that perform dimension reduction into 2D space, so it is likely that this algorithm introduces certain level of disentanglement.

[1] Eastwood, Cian, and Christopher KI Williams. "A framework for the quantitative evaluation of disentangled representations." 6th International Conference on Learning Representations. 2018.

[2] Kim, Hyunjik, and Andriy Mnih. "Disentangling by factorising." International conference on machine learning. PMLR, 2018.

**Questions:**

What is the reason that the authors choose VAEs and GANs as the two specific types of models that they want to implement the framework with? I mean my main concern is that, since disentangled representation learning has been studied for almost ten years from now, there have been a lot of variants of VAEs that were developed for learning disentangled representations. So I wonder 1) whether their framework can be applied to those variants of VAEs, and 2) if it is possible to compare against those models, such as TC-VAE[3], FactorVAE[4], and HFVAE[5]


[3] Chen, Ricky TQ, et al. "Isolating sources of disentanglement in variational autoencoders." Advances in neural information processing systems 31 (2018).

[4] Kim, Hyunjik, and Andriy Mnih. "Disentangling by factorising." International conference on machine learning. PMLR, 2018.

[5] Esmaeili, Babak, et al. "Structured disentangled representations." The 22nd International Conference on Artificial Intelligence and Statistics. PMLR, 2019.

---

> ### Author Response · Authors · 2024-11-22
> **Response to Reviewer 4KGP, Part 1**
>
> Dear 4KGP, thank you for taking the time and effort to review our paper, which helps us improve the depth of our work. Your valuable suggestions allowed us to present a more comprehensive evaluation and reinforce the generality of our framework. We deeply appreciate your insightful feedback and thoughtful comments.
>
>
> >W1: While they validated their framework on multiple datasets, they only provided qualitative results in terms of the disentanglement of learned representations. I would hope that they can evaluate the results more quantitatively and make comparisons based on metrics of disentanglement, e.g. the metrics used in [1] and [2]. I guess the reason I want to see this is it is difficult to see the disentanglement only based on t-SNE plots, because t-SNE itself will employ an algorithm that performs dimension reduction into 2D space, so it is likely that this algorithm introduces a certain level of disentanglement.
>
> **A1**: We appreciate your suggestion regarding the evaluation of disentanglement using quantitative metrics. In light of your suggestions, we have acknowledged the importance of quantitative evaluation for disentanglement and followed your recommendation to use metrics in [1]. Specifically, we have considered three other metrics: completeness, disentanglement score, and informativeness. Note that the bigger the values of completeness and disentangle score, the better the performance of disentanglement, and the smaller the values of informativeness, the better the performance of disentanglement. Since these metrics require the ground truth and estimated latent variables, we have conducted experiments on the synthetic datasets and compared the iMSDA [2] as follows:
>
>
> | MCC ↑   | CG-VAE | iMSDA |
> |-------------------|--------|-------|
> | Number of Domains  |                    |                 |
> | 2                 | 0.953 (0.007)      | 0.869 (0.002)   |
> | 4                 | 0.952 (0.017)      | 0.881 (0.043)   |
> | 6                 | 0.956 (0.016)      | 0.936 (0.025)   |
> | 8                 | 0.971 (0.009)      | 0.959 (0.017)   |
>
>
> | Completeness ↑   | CG-VAE | iMSDA |
> |-------------------|--------|-------|
> | Number of Domains  |        |       |
> | 2                 | 0.608 (0.115) |   0.405 (0.063)    |
> | 4                 | 0.632 (0.054) |   0.477 (0.028)    |
> | 6                 | 0.626 (0.138) |    0.663 (0.066)   |
> | 8                 | 0.729 (0.064) |    0.758 (0.042) |
>
> | Disentanglement Score ↑ | CG-VAE | iMSDA |
> |-------------------|--------|-------|
> | Number of Domains  |        |       |
> | 2                 | 0.586 (0.118) |   0.306 (0.032)   |
> | 4                 | 0.589 (0.076) |   0.498 (0.033)  |
> | 6                 | 0.611 (0.125) |    0.514 (0.045)  |
> | 8                 | 0.697 (0.019) |    0.710 (0.026)  |
>
> | Informativeness ↓  | CG-VAE | iMSDA |
> |-------------------|--------|-------|
> | Number of Domains  |        |       |
> | 2                 | 0.265 (0.018) |   0.317 (0.028)  |
> | 4                 | 0.256 (0.045) |    0.360 (0.045)  |
> | 6                 | 0.212 (0.016) |    0.282 (0.001) |
> | 8                 | 0.206 (0.016) |     0.225 (0.013)  |
>
>
>
>
> The results show that our method achieves good performance even when the number of domains is small, which verifies our theoretical results. In light of your suggestions, we have added the experiment results of completeness and the disentanglement score in Lines 432-444. And the experiment results of informativeness can be found in Lines 1250-1255.
>
> [1] Eastwood, Cian, and Christopher KI Williams. "A framework for the quantitative evaluation of disentangled representations." 6th International Conference on Learning Representations. 2018.
> [2] Li, Zijian, et al. "Subspace identification for multi-source domain adaptation." Advances in Neural Information Processing Systems 36 (2024)
>
> > W2.1(Q1): What is the reason that the authors choose VAEs and GANs as the two specific types of models that they want to implement the framework with?
>
>
> *A2.1**: This is a great observation. Although the proposed framework can be implemented by both VAEs and GANs, we learned from the results that they can facilitate different tasks. For instance, in order to discover learned factors, the latent representation in VAE can be immediately used. On the other hand, for the purpose of multi-domain image generation, where high-quality sample generation is critical, GAN is a particularly well-suited choice.
>
> In light of your questions, we have considered the GC-GAN and GC-VAE in the synthetic experiments, which are shown in Lines 390-395 and Lines 432-444. For real-world experiments, the results of these models can be found in Lines 445-450.

---

> ### Author Response · Authors · 2024-11-22
> **Response to Reviewer 4KGP, Part 2**
>
> > W2.2(Q2): I mean my main concern is that, since disentangled representation learning has been studied for almost ten years now, there have been a lot of variants of VAEs that were developed for learning disentangled representations. So I wonder 1) whether their framework can be applied to those variants of VAEs, and 2) if it is possible to compare against those models, such as TC-VAE[3], FactorVAE[4], and HFVAE[5].
>
> **A2.2**: Yes, you are right! Our method can surely be applied to other variants of VAE like TC-VAE, FactorVAE, and HFVAE. In light of your suggestion, we have extended our experiments to include these VAE variants and evaluated their performance using our framework on simulation datasets with different metrics mentioned in A1, which are shown as follows:
>
>
> | MCC ↑         | CG-VAE | TC-VAE | FactorVAE | HFVAE |
> |-------------------|--------|--------|-----------|-------|
> | Number of Domains  |        |        |           |       |
> | 2             | 0.953 (0.007)    | 0.946 (0.009)    | 0.882 (0.012)       | 0.944 (0.006)   |
> | 4             | 0.952 (0.017)    | 0.900 (0.012)    | 0.876 (0.004)       | 0.932 (0.008)   |
> | 6             | 0.956 (0.016)    | 0.919 (0.014)    | 0.837 (0.003)       | 0.907 (0.065)   |
> | 8             | 0.971 (0.009)    | 0.934 (0.008)    | 0.869 (0.013)       | 0.956 (0.017)   |
>
>
> | Completeness ↑   | CG-VAE | TC-VAE | FactorVAE | HFVAE |
> |-------------------|--------|--------|-----------|-------|
> | Number of Domains  |        |        |           |       |
> | 2                 | 0.608 (0.115) | 0.553 (0.095) | 0.814 (0.052) | 0.633 (0.051) |
> | 4                 | 0.632 (0.054) | 0.661 (0.179) | 0.732 (0.064) | 0.497 (0.031) |
> | 6                 | 0.626 (0.138) | 0.600 (0.068) | 0.626 (0.126) | 0.612 (0.099) |
> | 8                 | 0.729 (0.064) | 0.793 (0.059) | 0.590 (0.185) | 0.719 (0.074) |
>
> | Disentanglement ↑ | CG-VAE | TC-VAE | FactorVAE | HFVAE |
> |-------------------|--------|--------|-----------|-------|
> | Number of Domains  |        |        |           |       |
> | 2                 | 0.586 (0.118) | 0.499 (0.015) | 0.812 (0.057) | 0.502 (0.021) |
> | 4                 | 0.589 (0.076) | 0.673 (0.177) | 0.732 (0.062) | 0.494 (0.024) |
> | 6                 | 0.611 (0.125) | 0.637 (0.095) | 0.624 (0.120) | 0.568 (0.150) |
> | 8                 | 0.697 (0.019) | 0.697 (0.087) | 0.589 (0.195) | 0.729 (0.099) |
>
> | Informativeness ↓  | CG-VAE | TC-VAE | FactorVAE | HFVAE |
> |-------------------|--------|--------|-----------|-------|
> | Number of Domains  |        |        |           |       |
> | 2                 | 0.265 (0.018) | 0.266 (0.026) | 0.489 (0.046) | 0.266 (0.025) |
> | 4                 | 0.256 (0.045) | 0.430 (0.034) | 0.464 (0.005) | 0.276 (0.031) |
> | 6                 | 0.212 (0.016) | 0.392 (0.030) | 0.527 (0.013) | 0.294 (0.032) |
> | 8                 | 0.206 (0.016) | 0.336 (0.046) | 0.469 (0.058) | 0.227 (0.023) |
>
>
> We have added these experiments in Appendix E.1. The results validate that our method can be effectively extended to different VAE variants.

---

> ### Author Response · Authors · 2024-11-24
>
> Dear Reviewer  4KGP,
>
> We sincerely appreciate your taking the time to review our manuscript and providing us with your insightful questions and suggestions. As the rebuttal discussion period is limited, we eagerly await any additional feedback you may have. We are more than willing to have further discussions with you.
>
> Best regards,
>
> Authors of submission 4906

---

> > ### Comment · Reviewer_4KGP · 2024-11-27
> >
> > I really appreciate the detailed reply from the authors, as well as the extra experiments they did, considering the relatively short period before the rebuttal. It's great to see the comparsion across those variants of VAEs, which will give a more comprehensive evaluation. Overall, my questions have been addressed, and I decide to raise my score to 6.

---

> > > ### Author Response · Authors · 2024-11-27
> > >
> > > Dear Reviewer 4KGP,
> > >
> > > We are pleased to hear that your issues are addressed. Thank you for your support and recognition of our work.
> > >
> > > Best regards,
> > >
> > > Authors of submission 4906

---

### Official Review · Reviewer_Fw6r · 2024-11-04

**Soundness:** 3
**Presentation:** 2
**Contribution:** 2
**Rating:** 6
**Confidence:** 3

**Summary:**

A disentangled representation learning framework with two types of identifiability guarantees has been proposed. By forcing a sparse mixing procedure, the subspace identifiability is ensured, while sufficient changes in the distribution of latent variables promote componentwise identifiability of latent variables.

**Strengths:**

- Theoretical understanding of identifiability issue in the disentangled representation
- Proposing two generative implementations based on VAE and GAN using a domain encoding network and sparse mixing constraints
- Synthetic and real-world experiments

**Weaknesses:**

The notations are heavy and hard to follow. Please provide a comprehensive notation table at the beginning of the paper. and explicitly define each symbol when it's first introduced. For example,
- In equation (2), what is the meaning of two subscripts of $\mathbf{x}_{k, (1)}$.
- Other notation also needs to be defined carefully, $\backslash k$ (all index but $k$) ?
- In equation (6), it seems there is a missing integral (the second integral has been omitted).

There are English typos and errors, e.g., line 157 (contributions --> contribute)

In synthetic experiments, there is no comparison with any nonlinear ICA methods. Comparing the CG-VAE, and CG-GAn with one or two relevant nonlinear ICA methods can be beneficial for readers to understand the advantages of incorporating domain encoding network and sparse mixing constraints in the disentangled representation.

**Questions:**

- In line 220, the index in $\mathbf{z}_{o1}$ should be $o_1$ ?
- Can you provide some intuition why Sufficient Changes for Component-wise Identification require the second derivative assumption, while this is not the case for the subspace Identification? This could help readers better understand the theoretical foundations of the proposed method.

---

> ### Author Response · Authors · 2024-11-22
> **Response to Reviewer Fw6r, Part 1**
>
> Dear Reviewer Fw6r, thank you for your insightful and constructive review. Your comments have significantly helped us refine the clarity and readability of our work as well as the soundness of our experiments. We greatly appreciate the time and effort you dedicated to reviewing our submission.
>
> >W1: The notations are heavy and hard to follow. Please provide a comprehensive notation table at the beginning of the paper, and explicitly define each symbol when it's first introduced. For example,
> >- In equation (2), what is the meaning of two subscripts of $\mathbf{x}_{k}^{(1)}$?
> >- Other notation also needs to be defined carefully, $\backslash k$ (all index but $k$)?
> >- In equation (6), it seems there is a missing integral (the second integral has been omitted).
> >There are English typos and errors, e.g., line 157 (contributions --> contribute).
>
> **A1**: We appreciate your careful review and very helpful feedback. Below, we provided the point-to-point response to the comment and we have updated the paper accordingly.
> - We denote by $\mathcal{x} \in \mathbb{R}^n$ a vector of latent variables with the dimensionality of $n$. The subscript $k$ refers to a specific index or position of $\mathcal{x}$, indicating that $\mathcal{x} _k$ is the $k$-th dimension of $\mathcal{x}$. Importantly, $\mathcal{x}$ is a random vector, and each $\mathcal{x} _k$ is a scalar random variable corresponding to one dimension of $\mathcal{x}$. The subscript $(1)$ indicates a particular instance or realization of the random variable $\mathcal{x} _{k}$. Thus, $\mathcal{x} _{k, (1)}$ represents the first instance or realization of the random variable $\mathcal{x} _k$. Thanks for your question, we have provided a notation description on Page 2.
>
> - You are right. $\mathcal{x} _{\backslash k}$ denotes the random vector $\mathcal{x}$ with the $k$-th component excluded.
>
> - Thanks for pointing it out. To avoid confusion, we have added the integral in Equation (6), which can be found in Line 318.
>
> - Typos: Thanks for your careful review. We have gone through the whole paper and corrected the typos.
>
>
> >W2: In synthetic experiments, there is no comparison with any nonlinear ICA methods. Comparing the CG-VAE, and CG-GAN with one or two relevant nonlinear ICA methods can be beneficial for readers to understand the advantages of incorporating domain encoding network and sparse mixing constraints in the disentangled representation.
>
> **A2**: Thanks for your valuable comment, which helps make the experiments more solid. In light of your suggestion, we have considered two relevant nonlinear ICA baselines: iMSDA [1] and iVAE [2], which leverage domain index as auxiliary variables for identifiability. Experiment results can be found in Lines 390-395 of the revised version.
>
> Specifically, we generate synthetic data following the mixing process defined in Equation (1) and illustrated in Figure 7(a). Since our proposed methods and the baselines are based on variational autoencoder (VAE) structures, we ensure fairness by employing the same encoder and decoder architectures for all methods. Additionally, we use identical hyperparameters, such as the learning rate and the number of training epochs. We consider scenarios with 2, 4, 6, and 8 different domains. The experimental results are presented in the following table:
>
> | MCC ↑   | CG-VAE | iMSDA | iVAE  |
> |:-----------------:|:------:|:------:|:------:|
> | Number of Domains  |        |       |
> |         2          | 0.953 (0.007) | 0.869 (0.002) | 0.756 (0.105) |
> |         4          | 0.952 (0.017) | 0.881 (0.043) | 0.807 (0.060) |
> |         6          | 0.956 (0.016) | 0.936 (0.025) | 0.815 (0.029) |
> |         8          | 0.971 (0.009) | 0.959 (0.017) | 0.920 (0.003) |
>
>
>
>
>
>
> Experimental results demonstrate that when the number of domains is relatively small, our method achieves significantly higher MCC, which validates the theoretical results of our paper.
>
>
> [1] Kong, Lingjing, et al. "Partial disentanglement for domain adaptation." International conference on machine learning. PMLR, 2022.
> [2] Khemakhem, Ilyes, et al. "Variational autoencoders and nonlinear ica: A unifying framework." International conference on artificial intelligence and statistics. PMLR, 2020.
>
>
> >W3(Q1): In line 220, the index in $z_{\mathbf{o}_1}$ should be $\mathbf{o}_1$?
>
> **A3**: Thank you for your careful review. We have corrected the index in $z_{\mathbf{o}_1}$ to $\mathbf{o}_1$.

---

> ### Author Response · Authors · 2024-11-22
> **Response to Reviewer Fw6r, Part 2**
>
> >W4(Q2): Can you provide some intuition why Sufficient Changes for Component-wise Identification require the second derivative assumption, while this is not the case for the subspace Identification? This could help readers better understand the theoretical foundations of the proposed method.
>
> **A4:** Thanks for the great question! Subspace identification is weaker than component-wise identification, and hence naturally requires less information in the data generation process and distributions. Let us start with the subspace identifiability, suppose we have two sets of latent variables with changing distributions denoted by $\mathcal{z} _s$ and $\mathcal{z} _c$, respectively. And $\hat{\mathcal{z}} _s$ and $\hat{\mathcal{z}} _c$ are the corresponding estimated variables. Moreover, we let the $i$- and $j$- component of $\mathcal{z} _s$ and $\mathcal{z} _c$ be $z _s^i$ and $z _c^j$, respectively. In order to see whether $\mathcal{z} _s$ is identifiable up to a subspace, we just need to show whether $\frac{\partial z _{s}^i}{\partial \hat{z} _c^j}=0$. To this end, only the first-order derivative of the logarithmic density distribution is needed. However, for the purpose of component-wise identifiability, we have to make additional conditions, such as conditional independence between $z _s^{i}$ and $z_s^{j}$ given all the remaining components. These conditions impose stronger constraints compared to subspace identifiability. Mathematically, this conditional independence implies that the second-order cross derivative of the logarithmic density distribution with respect to $\hat{z} _{i}$ and $\hat{z} _{j}$ is zero, which involves second-order derivatives. In light of your treasured question, we have provided this intuition in Appendix D.2.

---

> ### Author Response · Authors · 2024-11-24
>
> Dear reviewer Fw6r,
>
>
> Thanks for the time you dedicated to carefully reviewing this paper. It would be highly appreciated if you let us know whether our responses properly address your concerns, despite your busy schedule. Thanks a lot!
>
> Best regards,
>
> Authors of submission 4906

---

> > ### Comment · Reviewer_Fw6r · 2024-11-25
> >
> > Thanks for addressing my comments. I'll keep my score at 6.

---

> > > ### Author Response · Authors · 2024-11-25
> > > **Many thanks!**
> > >
> > > Dear Reviewer Fw6r,
> > >
> > > Thanks a lot for being responsible and responsive!  If there were a way for you to update your rating or comments to reflect your feedback, it would be extremely appreciated.
> > >
> > > Best wishes,
> > >
> > > Authors of #4906

---

### Author Response · Authors · 2024-11-25
**We hope AC can encourage Reviewers 4KGP for discussion**

We thank the ACs for the time and efforts dedicated to taking care of our paper. Despite your very busy schedule, we hope you can help encourage Reviewers 4KGP for discussion.

We believe our modifications can address the concerns raised by Reviewers 4KGP well. As Reviewer qELo said, our modification can well solve his/her similar questions.

We kindly request you consider these points in your final decision.

Thank you once again.

With best regards,

Authors of submission 4906

---

### Author Response · Authors · 2024-11-30

Dear AC,

Thank you very much for the time and effort you dedicated to handling the submissions.  All reviewers responded to our response, which we highly appreciate.  We would like to bring your attention to Reviewer Fw6r, who ever said he/she would like to increase the rating to 7, but later deleted the message.  Perhaps it is because there is no rating of 7 as an option.  We hope you can take this into account when making your decision.

Many thanks,

Authors of submission 4906

---

### Meta-Review · Area_Chair_Xmk6 · 2024-12-17

**Metareview:**

This paper proposes a disentangled representation learning framework that combines two existing methods—sufficient changes in the latent variable distribution and sparse mixing procedures—to ensure identifiability in deep generative models like VAEs and GANs. The approach is validated with both synthetic and real-world experiments, showing that it leads to better latent identifiability.

Strengths: The paper presents a solid theoretical framework for identifiability in disentangled representation learning and provides clear mathematical derivations, supported by experimental results. The combination of sparse mixing and sufficient changes in latent distributions represents an innovative and effective approach to improving identifiability.

Weaknesses: The paper could benefit from clearer notation and definitions, as well as a more comprehensive experimental evaluation, including comparisons with other relevant models like nonlinear ICA methods. Additionally, while the results are promising, further quantitative analysis of disentanglement metrics would strengthen the conclusions. Most were addressed during the rebuttal.

Recommendation: I recommend this paper for acceptance, as it makes a meaningful contribution to the field of disentangled representation learning, though improvements in presentation and additional experimental comparisons would enhance its impact.

**Additional Comments On Reviewer Discussion:**

During the review period, the authors addressed several key points raised by reviewers. Reviewer Fw6r highlighted issues with unclear notation, which the authors resolved by adding a comprehensive notation table and correcting typos and a missing integral in Equation (6). The reviewer also pointed out the lack of comparisons with nonlinear ICA methods, which the authors addressed by adding comparisons with iMSDA and iVAE, showing the superior performance of CG-VAE, especially with small domain sizes. Reviewer 4KGP raised concerns about the lack of quantitative disentanglement metrics, which the authors addressed by incorporating completeness, disentanglement score, and informativeness metrics, demonstrating CG-VAE's superior disentanglement. The authors also responded to concerns about Theorem A3's readability, clarifying it by breaking it into shorter sentences. Additionally, they added baseline methods like CG-GAN and multiple others, strengthening the experimental validation, and included intuitive metrics like MSE and R^2 for a more thorough evaluation of the model. These revisions significantly improved the clarity, experimental rigor, and overall quality of the paper.

---

### Decision · Program_Chairs · 2025-01-22

Accept (Poster)